# Development and Validation of an Efficient MRI Radiomics Signature for Improving the Predictive Performance of 1p/19q Co-Deletion in Lower-Grade Gliomas

**DOI:** 10.3390/cancers13215398

**Published:** 2021-10-27

**Authors:** Quang-Hien Kha, Viet-Huan Le, Truong Nguyen Khanh Hung, Nguyen Quoc Khanh Le

**Affiliations:** 1International Master/Ph.D. Program in Medicine, College of Medicine, Taipei Medical University, Taipei 110, Taiwan; m142109004@tmu.edu.tw (Q.-H.K.); d142109009@tmu.edu.tw (V.-H.L.); d142108005@tmu.edu.tw (T.N.K.H.); 2Department of Thoracic Surgery, Khanh Hoa General Hospital, Nha Trang City 65000, Vietnam; 3Department of Orthopedic and Trauma, Cho Ray Hospital, Ho Chi Minh City 70000, Vietnam; 4Professional Master Program in Artificial Intelligence in Medicine, College of Medicine, Taipei Medical University, Taipei 106, Taiwan; 5Research Center for Artificial Intelligence in Medicine, Taipei Medical University, Taipei 106, Taiwan; 6Translational Imaging Research Center, Taipei Medical University Hospital, Taipei 110, Taiwan

**Keywords:** low-grade gliomas, radiogenomics, machine learning, chromosome 1p/19q codeletion, molecular subtype, wavelet transform, magnetic resonance imaging, precision medicine, computer aided diagnosis, decision making

## Abstract

**Simple Summary:**

Low-grade gliomas (LGG) with the 1p/19q co-deletion mutation have been proven to have a better survival prognosis and response to treatment than individuals without the mutation. Identifying this mutation has a vital role in managing LGG patients; however, the current diagnostic gold standard, including the brain-tissue biopsy or the surgical resection of the tumor, remains highly invasive and time-consuming. We proposed a model based on the eXtreme Gradient Boosting (XGBoost) classifier to detect 1p/19q co-deletion mutation using non-invasive medical images. The performance of our model achieved 87% and 82.8% accuracy on the training and external test set, respectively. Significantly, the prediction was based on only seven optimal wavelet radiomics features extracted from brain Magnetic Resonance (MR) images. We believe that this model can address clinicians in the rapid diagnosis of clinical 1p/19q co-deletion mutation, thereby improving the treatment prognosis of LGG patients.

**Abstract:**

The prognosis and treatment plans for patients diagnosed with low-grade gliomas (LGGs) may significantly be improved if there is evidence of chromosome 1p/19q co-deletion mutation. Many studies proved that the codeletion status of 1p/19q enhances the sensitivity of the tumor to different types of therapeutics. However, the current clinical gold standard of detecting this chromosomal mutation remains invasive and poses implicit risks to patients. Radiomics features derived from medical images have been used as a new approach for non-invasive diagnosis and clinical decisions. This study proposed an eXtreme Gradient Boosting (XGBoost)-based model to predict the 1p/19q codeletion status in a binary classification task. We trained our model on the public database extracted from The Cancer Imaging Archive (TCIA), including 159 LGG patients with 1p/19q co-deletion mutation status. The XGBoost was the baseline algorithm, and we combined the SHapley Additive exPlanations (SHAP) analysis to select the seven most optimal radiomics features to build the final predictive model. Our final model achieved an accuracy of 87% and 82.8% on the training set and external test set, respectively. With seven wavelet radiomics features, our XGBoost-based model can identify the 1p/19q codeletion status in LGG-diagnosed patients for better management and address the drawbacks of invasive gold-standard tests in clinical practice.

## 1. Introduction

Amongst tumors from the central nervous system (CNS) negatively affecting millions of patients worldwide regardless of their age or gender, brain cancers account for the highest prevalence of more than ninety percent [1]. Gliomas, which originate from astrocytes, oligodendrocytes, oligo-astrocytes, and glioneuronal cells, are the most common type of brain cancers [1]. According to the 2016 World Health Organization (WHO) classification of CNS tumors, gliomas could be divided into diffuse low-grade gliomas (LGGs) (i.e., grade-II and grade-III gliomas), and high-grade gliomas. The criterion of diagnosing LGGs, which make up 30 percent of gliomas [2,3], requires evidence of isocitrate dehydrogenase (IDH) 1 and 2 mutations, without or with the presence of 1p/19q codeletion. More specifically, the diagnosis of oligodendroglioma requires both IDH and 1p/19q mutation, whereas only the mutation of IDH is considered to diagnose diffuse astrocytoma [2,4,5]. Various studies indicated that the LGGs with 1p/19q codeletion were more indolent [6,7,8]. Patients diagnosed with 1p/19q codeletion mutation status LGG significantly had their survival time improved and were more sensitive to therapeutics in terms of chemotherapy and radiotherapy compared with those with 1p/19q non-deleted tumors [9,10,11,12,13]. Thus, the manipulation of 1p/19q co-deleted mutation and the early detection of this chromosomal abnormality among patients diagnosed with LGGs is highly appreciated, as this can facilitate short-term and long-term management.

The current gold-standard procedure of identifying 1p/19q codeletion clinically is the examination of the tumor’s histopathological sample. The pathology samples are collected via the brain-tissue biopsy or the surgical resection of the tumor, which contain many underlying risks [3,14,15,16]. Additionally, the results interpreted from the cytological analysis of LGGs lack critical information about genomic biomarkers or pathognomonic imaging changes of gliomas or other brain cancers [10,16].

The extraction of tumor properties from medical images (e.g., magnetic resonance imaging (MRI), computed tomography (CT), positron emission tomography (PET), etc.) is one of the various approaches conducted to address the limitations of mentioned methods [10,16,17]. Following the ubiquity of medical images, the term “radiomics” is developed and defined as quantifiable data or features derived from those images that can be utilized to uncover disease properties for diagnosis, making decisions, and further clinical evaluation of morbidities (e.g., tumors, lesions, etc.) [17,18,19,20,21].

Since the advent of precision and personalized medicine, machine learning (ML) has received great interest as a promising tool for diagnosing and predicting optimal treatment for cancer patients [3,22,23,24]. Recent ML models were proposed to assist the non-invasive detection of 1p/19q co-deleted LGGs, based on the intensity, texture, and geometry obtained via radiomics features from medical images [4,5,10,18,19,25,26,27,28,29,30,31,32,33]. However, the performance of most models remained unstable on the external test set, as the accuracy score ranged from 0.68 to 0.93. Additionally, the database used to create these models contained mostly retrospective cohorts, or they had to use many features for the prediction, and yet, the predictors could not replace the histopathological methods in the tumor genetic profiling.

In this study, we hypothesize that a ML model could predict the mutation of 1p/19q chromosomal arms using only a number of optimal radiomic features. We propose an eXtreme Gradient Boosting (XGBoost) model to tackle the limitations of most models in terms of prolonged runtime, stable performance on different data, and reproducibility under various conditions, which would then contribute to targeted decisions and reduce the adverse effects induced by invasive diagnostic methods on LGG patients. Both our training and validation database included preoperative patients from retrospective studies [10,34], and their condition of 1p/19q was verified by clinical gold standard examinations. With only seven optimization features, we believe that further studies can build on this framework to optimize cancer prediction models in the future. 

## 2. Results

The updated The Cancer Imaging Archive (TCIA) database of 159 LGG patients [10] was used as our training database. All MR images were downloaded via the National Biomedical Imaging Archive (NBIA) Data Retriever (version 3.6, https://wiki.cancerimagingarchive.net/display/NBIA/Downloading+Images+Using+the+NBIA+Data+Retriever, accessed on 20 March 2021). Statuses of 1p/19q chromosomal arms were fully reported based on the gold standard histopathology examination via brain biopsy or open surgical tumor resection.

### 2.1. Baseline Comparison among Different Machine Learning Algorithms and Selecting the Baseline Machine Learning Model

We obtained 159 complete region-of-interest (ROI) segmentations from the previous study by Akkus et al. [10,26]. Next, 851 radiomic features were derived from each segmentation via 3D Slicer Software (version 10.4.2) [35]. The lack of negative instances (or 1p/19q non-deleted mutations) may affect the performance of models (especially for specificity score). Therefore, the Synthetic Minority Over-sampling Technique (SMOTE) [36] would be applied in the following steps to handling the imbalance and improve the model’s specificity. Among the five machine learning algorithms (with default parameters) surveyed on the initial training set (159 instances and 851 radiomic features), the eXtreme Gradient Boosting (XGBoost) algorithm outperformed other classifier algorithms, as it correctly predicted 69.17% total instances using all features (Table 1). The accuracy score of Random Forest (RF) and Adaptive Boosting (AdaBoost) ranked second and third at 66.67% and 60.99%, respectively. On the other hand, the performance of k-Nearest Neighbors (kNN) and Logistic Regression (LR) algorithm were noticeably low. kNN observed the lowest accuracy score with 42.16%, whereas LR accurately predicted 47.8% total observations.

XGBoost also ranked first in sensitivity with 73.16%, precision with 81.19%. Even though the specificity was relatively low at 61.51%, it still outperformed the other algorithms. Moreover, XGBoost showed a significant performance in terms of AUC and AUPRC (0.71 and 0.827, respectively). In this experiment, XGBoost was therefore chosen to be our baseline algorithm, for feature selections with SHapley Additive exPlanations (SHAP) analysis [37] and further considerations.

### 2.2. Radiomics Signature Building

We performed SMOTE on the dataset with 159 instances and 851 original features before features selection. In this study, we performed two steps of features selection using Spearman’s Correlation Coefficient (SCC) [38] and the combination of XGBoost-SHAP analysis. First, with the threshold of SCC equal to 0.8, we kept features with SCC greater than 0.8 and discarded features with at least one SCC less than 0.8. As a result, 68 features were removed.

Then, features selection continued with the combination of XGBoost-SHAP analysis. Only 37 features were retained after this step. The SHAP values bee swarm plot expressed the SHAP values of the 20 best features and ranked them by their importance (out of 37 retained features) (Figure 1). In addition, the below part of Table 1 shows that the performance was increased when we only used our 37 features inserting into our model.

### 2.3. Model Ensembling and Predictions on the Training Set

Finally, we added each feature from the 37 aforementioned features to the XGBoost-baseline model to seek the model with the best combination of XGBoost algorithm and certain set of features. We found the model with the highest accuracy score at 87% with the first seven features and the XGBoost algorithm with the following parameters: max_depth = 6; min_child_weight = 1; max_delta_step = 0; lambda = 1; alpha = 0; scale_pos_weight = 1 (Figure 2). Names of the chosen radiomic features were provided in Table 2.

The original first-order skewness was the best radiomic feature in our study. Skewness of the first-order matrix refers to the measurement of figure distribution’s symmetry around the Mean value (i.e., the average gray level of intensity that lies within the ROI). Depending on the symmetry, this feature could be positive or negative.

Our final model, build on XGBoost algorithm (max_depth = 6; min_child_weight = 1; max_delta_step = 0; lambda = 1; alpha = 0; scale_pos_weight = 1) and seven optimal features (Table 2), was applied to predict the LGG 1p/19q codeletion mutations on the training set, with five-fold cross-validation. The results were promising, with 87% in accuracy, 88.2% in sensitivity, and 77.2% in specificity. Our model achieved an AUC of 0.87, and we also plotted the area under a receiver operating characteristics (AUC ROC) curve and the precision–recall curve to visualize the performance of our model in each fold (Figure 3).

### 2.4. Comparison with Previous Studies on 1p/19q Status Prediction

LGG gliomas are malignant tumors; however, the detection of 1p/19q codeletion mutation could enhance the prognosis and the therapeutic procedures [9,10,11,12,13]. Various studies were conducted to address the limitations in early diagnosing 1p/19q mutated LGG gliomas. We compared concretely our proposed model with models formerly developed by other authors in Table 3. It was notable that the deep learning algorithm-based models were superior to our model, whereas the XGBoost-7 outperformed a variety of the ML frameworks.

### 2.5. Performance Results on the External Test Set

We applied our optimal model to the external test set with unseen data. The dataset contained 65 LGG patients with 52 1p/19q non-deleted and 13 1p/19q codeleted mutations. Our model accurately classified 82.80% of 1p/19q codeleted LGG, with 94.10% in specificity and 33.30% in sensitivity.

### 2.6. Performance Results of Our Radiomics Model on Different Patient Subgroups

We also used the model to predict 1p/19q codeletion mutations in different patient groups according to grades and types of LGG cells in the training and external test set (Table 4).

The accuracy scores were not lower than 64% in two sets. In particular, the model’s predictive ability achieved high accuracy for LGG grade 2, LGG type Astrocytoma, and Oligoastrocytoma in the training set and LGG grade 3 in the test set, with AUC reaching 0.876, 0.85, 0.836, and 0.847, respectively. Moreover, the model’s ability to recognize codeletion mutations was considerable for LGG grade 2, 3, or LGG type Oligoastrocytoma and Oligodendroglioma (with high sensitivity over 80% for these grades and types).

## 3. Discussion

In this study, we proposed a predictive model based on XGBoost algorithm and seven radiomic features. Our training database contained 159 LGG patients with semi-automated segmentations by Akkus et al. [26]. The authors’ method fully segmented tumors (or ROIs); however, there was still the possibility of errors due to user-to-user variability. A study by Rundo et al. [40] performed semi-automatic segmentation of brain tumors based on the presence of a cellular automata model and Gamma Knife. The method proposed by Rundo et al. was effective in brain tumor recognition and reproducible. However, auto-segmentation can reduce the time for segmentation while minimizing human error. Pereira et al. [41] proposed an MRI-based auto-segmentation method for patients with gliomas. In the future, we will apply the auto-segmentation technique in further studies and take advantage of the development of deep learning algorithms in identifying cancer mutations for better management of cancer patients.

Many studies have used machine learning or deep learning to identify 1p/19q codeletion mutations in LGG patients based on medical imaging. However, some studies still need to use many radiomic features [33], or the model’s performance is not high [28,30,39]. On the other hand, our model achieved a high performance using only seven optimal radiomic biomarkers. We compared our model’s performance with previously developed ML models by other authors. The similar XGBoost-based model by Shboul et al. [33] achieved the AUC, sensitivity, and specificity of 0.8 ± 0.04, 0.75 ± 0.08, and 0.85 ± 0.06, respectively. Given that the total features sorted out by Shboul et al. [33] were more than ours (15 vs. 7), our model achieved a higher score of AUC (0.8 vs. 0.85), and sensitivity (0.75 vs. 0.88), but lower specificity (0.85 vs. 0.77). Furthermore, our model outperformed the SVM-based model proposed by van der Voort et al. [39] (AUC 0.723 vs. 0.85, sensitivity 0.732 vs. 0.88, specificity 0.617 vs. 0.77), and Rathore et al. [30] (accuracy 75.15% vs. 86.80%, sensitivity 0.82 vs. 0.88, specificity 0.74 vs. 0.77). A two-step classification based on the presence of the T2-FLAIR mismatch sign yielded by Batchala et al. [31] in 2019 surpassed our results regarding the training process, but overall lower scores were observed in the test set (accuracy ranged from 79.2% to 81.1% vs. 82.80%). The authors also included the contribution of humans in the procedure; this was appreciated. However, it may be prone to inconvenience and excessive total cost of the implementation in real-world conditions. Since we used fewer features than previous studies and our wavelet features were not hard to be produced, clinicians may actually use our model to predict 1p/19q mutation status with less time and more accuracy than other studies.

Referring to the deep learning models, the compared results were also eminently favorable. In 2019, a review article by Kocak et al. [27] yielded that the model could correctly classified the 1p/19q deletion status at an average AUC of 0.869, which was somewhat lower than our XGBoost model’s performance (0.869 vs. 0.85). Matsui et al. [28], in 2016, introduced a comprehensive model to predict the mutation of concerned chromosomal arms, and achieved only 58.5% accuracy on MRI database, despite the reasonable results on the combination of different forms of medical images. Nonetheless, there were various remarkable performances via the application of deep neural networks. In 2017, the performance of a convolutional neural network (CNN)-based model by Akkus et al. [10] generated an accuracy of 87.7% on the same dataset used in this study.

Yogananda et al. [5], in 2020, proposed a network correctly predicted the 1p/19q co-deletion with a prominent accuracy level of 93.46%, compared with 86.80% obtained from our research. The difference lies in the fact that the author has used deep learning algorithms to learn and predict. Deep learning algorithms have been proven highly effective in auto-segmentation, feature extraction, and learning capacity. However, training a deep learning model requires big data, massive time, and hardware. In contrast, our current model used a machine-learning algorithm (XGBoost), which requires less data and reduces the runtime to complete. Our model’s highly accurate prediction had clinical significance as it significantly reduced the time for diagnosing the 1p/19q codeletion mutation to adjust the treatment plan accordingly. Moreover, using only a few radiomics features could reduce the complexity of the diagnostic process but still ensures the model’s performance.

Shortening the time to detect the type of 1p/19q mutation in patients with LGG is important in terms of the treatment and prognosis of LGG. In clinical practice, it is necessary to shorten the runtime of the model, while ensuring high accuracy in predictions. Our model achieved high performance and stability based on seven optimal features with a ten-second runtime for MRI images with tumor segmentation. Problems can arise during the extraction of radiomics features from tumor images. In this study, the time from extracting radiomics features until the prediction was made was 2.8 + −0.3 min, achieving AUC 0.85 + −0.06. The deep learning model proposed by Yogananda et al., with a runtime of three minutes (from tumor segmentation to results), achieved the AUC of 0.95 + −0.01 in predictions. Despite the similar runtime, our model had a lower predictive performance compared with the model proposed by Yogananda et al. In the future, we will focus on improving the model’s performance while maintaining or further shortening the total time of the process from image input to output.

In terms of the performance of our model on the external test set, the sensitivity was low with 33.30%, compared with 94.10% of specificity. The overall accuracy reached 82.80%, which indicated that the performance was reasonable. However, the imbalance of sensitivity and specificity was due to the imbalance of the external test set, with 52 1p/19q non-deleted LGG patients (accounting for 80% of the data) and only 13 1p/19q codeletion LGG patients (accounting for 20% of the data). This could be interpreted as our model can identify accurately LGG patients with 1p/19q non-deleted status to apply appropriate cancer therapies. It is necessary to evaluate our model on different external test sets with more balanced data for further assessment and to validate the model’s sensitivity. Moreover, our model can be applied into cross-group training of LGG patients. As shown in Table 5, we reached promising performance when training with different grades, or tumor subtypes. Especially, our model worked well on oligoastrocytoma and oligodendroglioma subtypes with high sensitivity even though we had a small sample size in each problem.

Wavelet features made up the majority of the most important features in our study. Many studies have shown the strengths of wavelet features in image compression and preprocessing and use these features for classification [42,43,44]. Although no study has clearly shown the wavelet features’ advantage over the remaining features, the aforementioned advantages make wavelet features important in predicting based on medical imaging.

To the best of our knowledge, in the field of identifying the LGG 1p/19q codeletion status, our XGBoost-7 was the first model that contained the least number of involved radiomic features, and still yielded reasonable results against the training and external test set. This appealing finding indicated a prospect, in which the physicians would be capable of detecting the LGG with 1p/19q codeletion by T1- or T2-weighted MR images and only seven radiomic features, without the need of invasive biopsy or tumor resection. Patients would receive suitable management plans as quickly as possible, hence attenuating the risks of treatment-induced adverse effects and unnecessary medical procedures.

However, some limitations still need to be addressed. First, the modest datasets used for training and validating tasks conflicted with the real-world situations, as a large number of LGG patients with known 1p/19q codeletion were not included. For the next steps, data of LGG patients from medical centers would be included to our trained model for evaluation. Second, the binary classification of LGG patients was insufficient, since this study lacked the crucial prognostic marker IDH 1 and 2 mutation [4,5]. In the future, we would conduct the multiclassification to contribute to preoperative treatment plans. Third, only MR images in the axial plane were considered. This unintendedly overlooked some potential characteristics of the tumor. For the next project, the authors would try to propose a model that is compatible with all MRI planes, thence comprehensively predict the 3D shape of the tumor. Fourth, also a constraint of most studies, only a retrospective database was considered; hence, the application on prospective cohorts would gauge the robustness of our model more completely. Finally, despite the stability and promising results of XGBoost algorithm towards the unbalanced database [45,46], and the given result on the training and external validation cohort, the above citations implied that the combination of the present model and deep learning algorithms [47] may significantly enhance the outcomes in the future.

## 4. Materials and Methods

Our workflow (which was exhibited in Figure 4), from data extraction to the external validation, comprised three main steps: (I) Data extraction from two public databases from former articles [10,34]: one database was used for training and another one was used as the external test set; (II) radiomic feature extraction and feature selection using SCC, XGBoost, and SHAP analysis; (III) applying the ML algorithm on refined features to predict the 1p/19q codeletion status of LGG patients on the training and external test set.

### 4.1. Patient Cohort

For the training database, we obtained 159 LGG patients (WHO Grade II and III) with confirmed preoperative diagnosis, histopathological result of LGG, 1p/19q mutation status, and complete region-of-interest (ROI) segmentation of tumors in three axial slices in NiFTI format from the database published on The Cancer Imaging Archive (TCIA) by Akkus et al. [10,48]. The National Biomedical Imaging Archive (NBIA) Retriever version 3.6, which is an open-source software for downloading medical images from TCIA, was used to download and extract the data. There were 57 patients with non-deleted 1p/19q condition and 102 co-deleted patients. Among the 102 co-deleted patients, there were 66 grade-II and 36 grade-III patients, compared with 38 and 19 patients in the respective grades of the non-deleted group. In terms of the tumor types, i.e., astrocytoma, oligo-astrocytoma, and oligodendroglioma, the number of patients diagnosed with oligo-astrocytoma accounted for the highest proportion of 61% regardless of the 1p/19q mutated condition, whereas people suffering from astrocytoma made up only 10.7% of the total patients.

For the external test set, we recruited 65 LGG patients from the previous study by Bakas et al. [34], which was published on TCIA [48]. The 1p/19q mutation status of all patients was confirmed by histopathological samples. In detail, 6 of the 13 1p/19q-co-deleted and 22 of the 52 1p/19q-non-deleted patients were WHO-Grade II LGGs, and all of the 13 1p/19q-co-deleted LGGs were oligodendrogliomas. The external test cohort was collected using the same steps as the training cohort with the support of the NBIA Retriever. Detailed information of patients was shown in Table 5.

### 4.2. Feature Extraction

We used 3D Slicer software [35] (version 4.10.2; released on 10 October 2012; last updated on May 17, 2019) to extract radiomic features from the MR images with ROI segmentations. We installed seven 3D Slicer extensions for the radiomic feature extractions, including SlicerRadiomics (https://github.com/AIM-Harvard/SlicerRadiomics, accessed on 20 March 2021) integrates PyRadiomics library [49] in 3D Slicer, supports calculation and extraction of radiomic features, DCMQI [50], PETDICOMExtension [51], QuantitativeReporting [50] (https://github.com/QIICR/QuantitativeReporting, accessed on 20 March 2021), SlicerDevelopmentToolbox (https://github.com/QIICR/SlicerDevelopmentToolbox*,* accessed on 20 March 2021), and SlicerRT [52].

Initially, the MR records were orderly imported to 3D Slicer for radiomic feature extraction. A total of 851 features derived from each record were afterwards stored in .tsv-formatted files. The features were classified into nine categories, e.g., original, wavelet high-high-high (HHH), wavelet high-high-low (HHL), wavelet HLH, wavelet HLL, wavelet LHH, wavelet LHL, wavelet LLH, and wavelet LLL. Each category comprised six sub-categories, namely first-order, Gray Level Co-occurrence Matrix (GLCM), Gray Level Size Zone (GLSZM), Gray Level Run Length Matrix (GLRLM), Neighbouring Gray Tone Difference Matrix (NGTDM), and Gray Level Dependence Matrix (GLDM), except for the original radiomics category with one sub-category (Shape) more than the others. However, in this study, we classified the features into four categories according to Aerts et al. [9], including Tumor Shape (containing 14 original shape features), Tumor Intensity (containing 18 original first-order features), Textures (containing 75 GLCM, GLSZM, GLRLM, NGTDM, and GLDM features), and 744 Wavelet features. The information about the radiomics classes was concretely described by Zwanenburg et al. [53] and Van Griethuysen et al. [49]

### 4.3. Data Mining

#### 4.3.1. Determining the Ground-Truth Labels

After the feature extraction step, our training database contained 159 samples with 851 features. For the ground-truth labels, we labelled “*n*/*n*” (i.e., 1p/19q non-codeletion status) as “0”, and “d/d” (i.e., 1p/19q codeletion status) as “1”.

#### 4.3.2. Select the Baseline Machine Learning Models

To evaluate which algorithms worked well on these radiomics features, we applied five ML algorithms to the training database. The ML algorithms included Logistic Regression (LR), k-Nearest Neighbors (kNN), Random Forest (RF), Adaptive Boosting (AdaBoost), and eXtreme Gradient Boosting (XGBoost). All ML algorithms were implemented using scikit-learn package in Python. All five algorithms were used inside grid search techniques (GridSearchCV method) to find the optimal parameters for preliminary assessment.

##### LR

LR is described as an algorithm used for calculating the correlations between independent variables toward one dependent variable, and the trendy prediction of related dependent values based on known independent inputs [40]. This algorithm can be divided into simple (SLR) and multiple LR (MLR). In this study, we applied MLR analysis to find whether there is a linear correlation between radiomic features and corresponding 1p/19q deletion status.

The following pattern exhibits the SLR equation:$$y={\;\beta }_{o}+\beta_{1}X_{1}+{\;\beta }_{2}X_{2}+\ldots +{\;\beta }_{n}X_{n}+\varepsilon$$
y, β_o_, β_1_…β_n_, X_1_…X_n_, and ε are denoted as the dependent variable, the intercept constant, the slope (also known as the regression coefficient), the independent variables (the radiomic features), and the random error, respectively.

##### kNN

Evelyn Fix and Joseph Hodges introduced an algorithm called “k-Nearest Neighbors” in 1951 [54]. Its main principle based on the intuitive assumption that nearby cases usually have many similarities. In this study, kNN was used as a supervised ML classifier. It appraised which class each neighbor was in and calculated the most preferred class within k neighbors of the concerned observation. Further details about kNN could be found at [55].

##### RF

The RF classifier algorithm is the combination of various decision trees where each tree would play a role in voting for the most popular class. Eventually, each separated input data would be classified into its relative class [56]. This algorithm, specifically, could be sufficiently used in different tasks including multiple or binary classification, and regression analysis. RF, in this study, was used to solve the binary classification problem on detecting 1p/19q mutation.

##### AdaBoost

AdaBoost is a boosting algorithm, which integrates the downgrade components into one robust model. The weight assignations to each training sample and classifier are implemented, i.e., the weaker the item and the stronger the classifier, the higher weight is added, to make it contribute more significantly to the outcome. A systematic review by Ying et al. [43] revealed more information about AdaBoost, its contribution and drawbacks in the field of high-yield ML. Recently, numerous projects have exploited AdaBoost and other boosting algorithms to build models used for radiomics prediction.

##### XGBoost

Similar to AdaBoost, XGBoost is the application of a sufficient gradient boosting algorithm, and hence, it becomes a favored choice in the midst of boosting techniques for its superior predictive performance, exact classification, and the capacity of administering imbalanced data. To be more particular, L1 and L2 regularization are also included in this algorithm, which are responsible for handling sparseness and attenuating overfitting. Moreover, the clinical records, e.g., the database of LGG patients used in this project, usually experience discrepancy data, which can be addressed using XGBoost. The detailed equation of XGBoost and its applications could be found in [33,57].

### 4.4. Handling the Imbalance between Two Classes and Features Selection Using the Spearman’s Correlation Coefficient (SCC) and SHAP Analysis

Prior to features selection, it was necessary to address the data imbalance. We used the Synthetic Minority Over-sampling Technique (SMOTE) [36]. SMOTE helps select minority instances from each class from the dataset and generate a synthetic point between the instances to make the classes more balanced [36]. In this study, SMOTE can address the low specificity on the training set.

We implemented the first step of radiomic feature selection by using the SCC. The pair-wise correlation coefficient between each pair of radiomic features was calculated. In this experiment, 0.8 was the threshold for determining if one feature would be excluded or included. Every pair, of which the SCC exceeded 0.8, was retained, whereas the SCCs of below 0.8 were prone to the exclusion. From 851 features, 68 features were filtered out, and leaving 783 features for further considerations. Moreover, SHAP analysis and RFE were integrated to find out the optimal 7 features. Our 7 features can be found in Table 2 for reproducing the methods.

### 4.5. Statistical Analysis

Statistical analysis was performed using Python. Our model is trained using 5-fold cross validation method, and then it is applied on an external validation set. Measurement metrics included Sensitivity, Specificity, Accuracy, ROC curve, and AUC.

Sensitivity is defined as the percentage of the 1p/19q co-deleted patients correctly predicted by the model on total patients with 1p/19q codeletion.
$$\mathrm{Sensitivity}\;=\frac{\mathrm{True}\;\mathrm{Positive}}{\mathrm{True}\;\mathrm{Positive}+\mathrm{False}\;\mathrm{Negative}}$$

Specificity is calculated by the ratio of patients with 1p/19q non-deleted patients recognized by the model and all 1p/19q non-deleted patients.
$$\mathrm{Specificity}\;=\frac{\mathrm{True}\;\mathrm{Negative}}{\mathrm{True}\;\mathrm{Negative}+\mathrm{False}\;\mathrm{Positive}}$$

Accuracy of a model is validated by the number of its correct predictions dividing by the whole number of involved patients.
$$\mathrm{Accuracy}\;=\frac{\mathrm{True}\;\mathrm{Positive}+\mathrm{True}\;\mathrm{Negative}}{\mathrm{Total}\;\mathrm{patients}}$$

The ROC–AUC and precision–recall (PR) curves for each fold in five-fold cross validation were plotted to visualize the overall performances of each algorithm.

## 5. Conclusions

We proposed an XGBoost-based model with solely seven wavelet radiomic features, to implement the classification of LGG 1p/19q codeletion status in the cohort of LGG patients, which was believed to contribute to the non-invasive diagnosis, individual cancer therapy, and long-term management for patients diagnosed with LGG. In the future, we intend to apply the model to multicenter data with many patients to evaluate the predictive power of the model. Additionally, we plan to conduct our experiments with auto-segmentation, using the state-of-the-art deep learning (convolutional neural networks, recurrent neural networks, etc.). Hopefully, our future work can contribute more to the early diagnosis of LGG cancer with 1p/19q codeletion mutation in clinical practice.

## Figures and Tables

**Figure 1 cancers-13-05398-f001:**
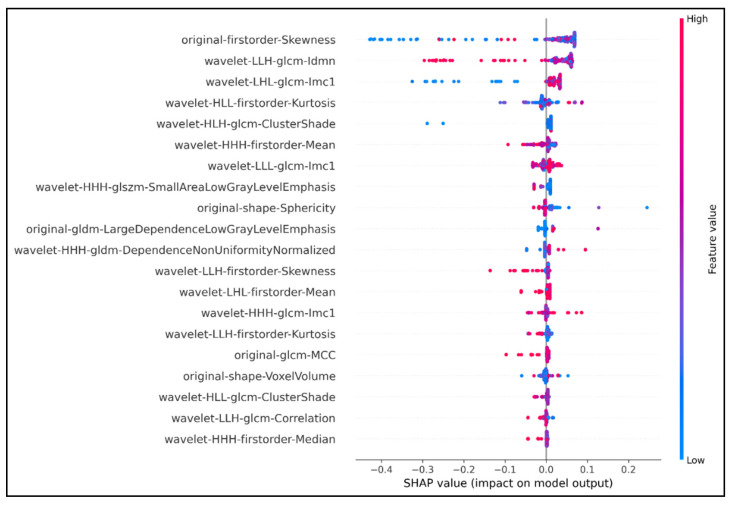
Top 20 features extracted using SHAP analysis. The horizontal axis showed how much the features contribute to the output predictor variable via SHAP values. Features were ranked based on their contribution to the output predictor; the higher the SHAP value, the higher the rank of the corresponding feature. The vertical axis with the color ranged from blue to red represented the value of a feature from low to high, respectively. The density of the colored dots in any SHAP value represented the strength or weakness of the feature in that SHAP value range. The original first order Skewness feature had the greatest impact on the predictability of the model. The SHAP value of original first order Skewness was 0.48 (absolute value of −0.4 plus +0.08). The density of red and purple points concentrated in the range from 0 to +0.08 indicated that the predictive value of this feature was highest as its corresponding SHAP value was from 0 to +0.08. On the other hand, the feature wavelet HHH first order Median ranked last in the graph, with the lowest SHAP value 0.06 (absolute value −0.05 plus +0.01), although it was still a relatively important feature (density thick red and purple dots).

**Figure 2 cancers-13-05398-f002:**
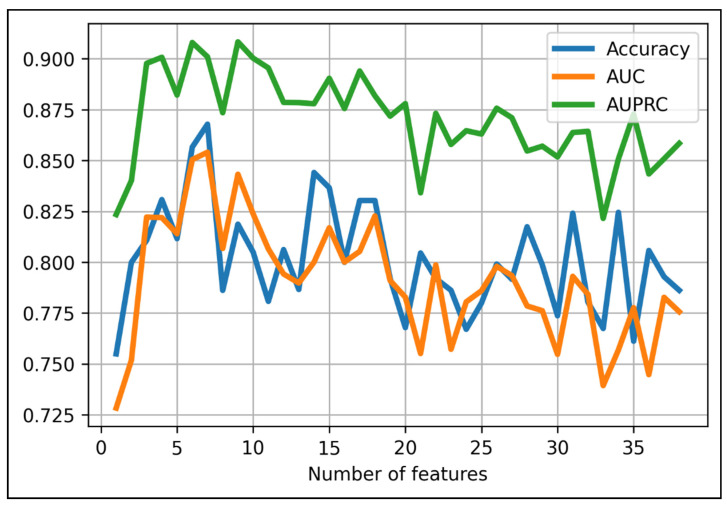
Performance results when using different numbers of top-rank features in predicting 1p/19 status using XGBoost algorithm. The optimal cut-off point belongs to the first seven features (accuracy = 87%).

**Figure 3 cancers-13-05398-f003:**
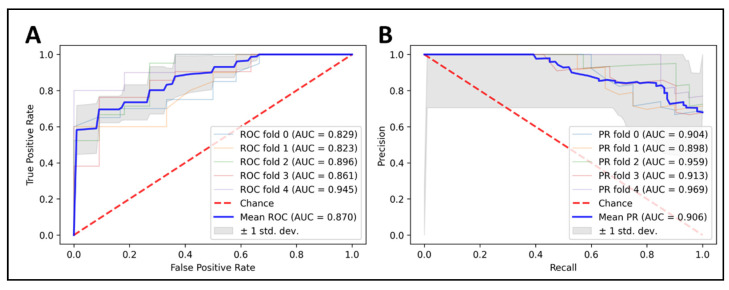
Predicting 1p/19q status using XGBoost classifier on top seven radiomics features. Cross-validation is implemented to get see the model consistency. (**A**) ROC curve analysis (AUC = 0.87), (**B**) PRC curve analysis (AUC = 0.906).

**Figure 4 cancers-13-05398-f004:**
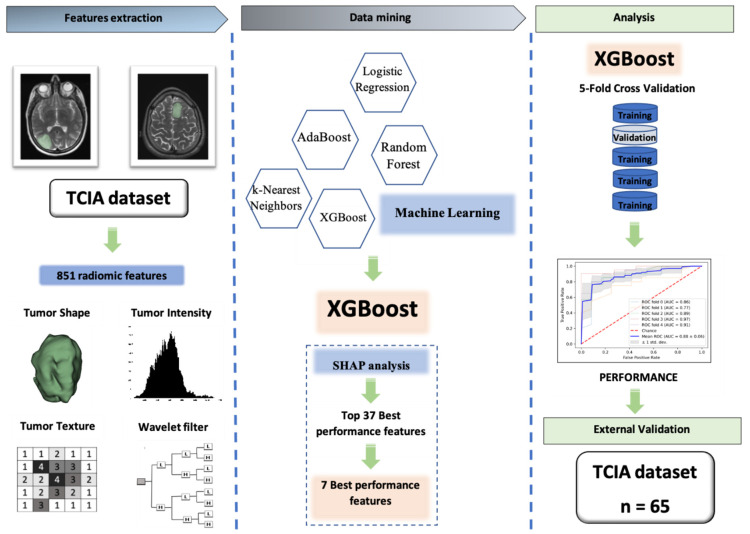
The workflow of our study. Firstly, data was extracted from two public database from former articles, one database was used for training and another one was used as the external test set; secondly, radiomic feature extraction and feature selection were performed using SCC, XGBoost, and SHAP analysis; finally, machine learning algorithm XGBoost classifier was applied on refined features to predict the 1p/19q codeletion status of LGG patients on training and external test set.

**Table 1 cancers-13-05398-t001:** Comparative performance among different machine learning algorithms using all radiomics features and top 37 features. The experiments were done and reported via 5-fold cross validation and SMOTE technique.

Features	Algorithm	Sensitivity	Specificity	Precision	Accuracy	AUC	AUPRC
All features	Logistic Regression	61.75 ± 5.17	32.26 ± 6.69	50.19 ± 9.31	47.80 ± 5.38	0.510 ± 0.095	0.687 ± 0.064
	k-Nearest Neighbors	56.20 ± 3.87	27.97 ± 3.46	44.19 ± 4.98	42.16 ± 3.93	0.434 ± 0.065	0.607 ± 0.027
	Random Forest	72.13 ± 3.07	57.88 ± 7.77	79.48 ± 12.34	66.67 ± 3.09	0.685 ± 0.061	0.799 ± 0.040
	AdaBoost	68.47 ± 7.31	47.03 ± 15.80	71.33 ± 11.39	60.99 ± 10.44	0.599 ± 0.082	0.740 ± 0.051
	XGBoost	73.16 ± 2.84	61.51 ± 12.25	81.19 ± 9.83	69.17 ± 6.09	0.710 ± 0.079	0.827 ± 0.057
37 features	Logistic Regression	64.95 ± 12.05	35.71 ± 6.79	44.10 ± 8.15	48.49 ± 8.20	0.587 ± 0.106	0.706 ± 0.091
	k-Nearest Neighbors	69.82 ± 8.04	42.83 ± 13.25	62.86 ± 8.61	58.55 ± 9.76	0.590 ± 0.116	0.720 ± 0.077
	Random Forest	74.10 ± 7.60	54.16 ± 10.94	75.29 ± 11.18	66.61 ± 7.57	0.713 ± 0.078	0.858 ± 0.041
	AdaBoost	71.80 ± 7.50	46.67 ± 6.67	70.67 ± 7.94	62.28 ± 4.72	0.632 ± 0.099	0.755 ± 0.090
	XGBoost	77.89 ± 5.75	55.85 ± 7.46	73.52 ± 3.96	69.21 ± 4.83	0.753 ± 0.058	0.809 ± 0.045

**Table 2 cancers-13-05398-t002:** Seven radiomics features using our final model.

Form	Type	Matrix	Name
original		First Order	Skewness
Wavelet	LLH	GLCM	Inverse Difference Moment Normalized (IDMN)
Wavelet	LHL	GLCM	Informational Measure of Correlation 1 (IMC1)
Wavelet	HLL	First Order	Kurtosis
Wavelet	HLH	GLCM	Cluster Shade
Wavelet	HHH	First Order	Mean
Wavelet	LLL	GLCM	IMC1

**Table 3 cancers-13-05398-t003:** Performance comparisons between our XGBoost-7 and other typical models.

Study	Algorithm	Dataset	Accuracy	Sensitivity	Specificity	AUC
van der Voort et al.	SVM	LGG 284	69.8	65.7	72.1	0.755
		LGG 108 [39]	69.3	73.2	61.7	0.723
Yogananda et al.	Deep learning	LGG 268 [5]	93.5	0.9	95	95.3
Akkus et al.	CNN	LGG 159 [10]	87.7	93.3	82.2	-
Shboul et al.	XGBoost	LGG 81	-	78	83	0.83
		LGG 23 [33]	-	75	85	0.8
Ours	XGBoost	LGG 159 [10]	87.0	88.2	77.2	0.87

**Table 4 cancers-13-05398-t004:** Performance results of our model on different patient subgroups. The final model used XGBoost algorithm and top seven features. The results show that we had promising performance on cross-grade and cross-tumor subtype problems.

Subtype		Training Data				External Validation Data					
		Acc	Sens	Spec	AUC	AUPRC	Acc	Sens	Spec	AUC	AUPRC
Grade	2	83.7 ± 11.7	87.9 ± 6.6	76.3 ± 22.4	0.876 ± 0.11	0.891 ± 0.07	82.1	16.7	100.0	0.695	0.855
	3	80 ± 15.1	97.2 ± 25.9	47.4 ± 4.1	0.759 ± 0.20	0.815 ± 0.13	88.6	50.0	96.6	0.847	0.885
Type	Astrocytoma	76.5 ± 8.6	25 ± 9.0	92.3 ± 11.9	0.85 ± 0.09	0.785 ± 0.078	-	-	-	-	-
	Oligoastrocytoma	80.4 ± 2.0	82.1 ± 0.9	78 ± 4.0	0.836 ± 0.04	0.858 ± 0.05	-	-	-	-	-
	Oligodendroglioma	93.3 ± 6.9	100.0 ± 0	0.0 ± 0	0.726 ± 0.07	0.864 ± 0.05	64.0	58.3	69.2	0.733	0.828

-: there is no ‘d/d’ class in this group.

**Table 5 cancers-13-05398-t005:** Patients’ characteristics.

Feature	Subtype	Training Cohort		External Test Cohort	
		d/d (*n* = 102)	*n*/*n* (*n* = 57)	d/d (*n* = 13)	*n*/*n* (*n* = 52)
Grade	2	66 (64.7%)	38 (66.7%)	6 (46.2%)	22 (42.3%)
	3	36 (35.3%)	19 (33.3%)	7 (53.8%)	30 (57.7%)
Type	Astrocytoma	4 (3.9%)	13 (22.8%)	0 (0%)	21 (40.4%)
	Oligoastrocytoma	56 (54.9%)	41 (71.9%)	0 (0%)	18 (34.6%)
	Oligodendroglioma	42 (41.2%)	3 (5.3%)	13 (100%)	13 (25%)

“d/d” means 1p and 19q are co-deleted, “*n*/*n*” means neither 1p nor 19q were deleted.

## Data Availability

Our source codes and data are freely available at https://github.com/khanhlee/LGG-radiomics, accessed on 20 March 2021.

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
