# Peer review of "Development and Validation of an Efficient MRI Radiomics Signature for Improving the Predictive Performance of 1p/19q Co-Deletion in Lower-Grade Gliomas"

_cancers, 2021, doi:10.3390/cancers13215398_

Round 1

Reviewer 1 Report

The authors present an interesting approach for the non-invasive, MR imaging-based detection of 1p/19q codeletion in low grade glioma. Despite the clinical motivation for this task, it should be stressed that neither the application, nor the presented model are novel and there is hence limited novelty in the presented approach. It may, however, still be an interesting additional contribution for the community.

Overall, the manuscript requires significant revision of the English language, phrasing and structuring. It was hard to follow sentence structure at times and there was no consistent separation between introduction, methods, results and discussion. At times, the phrasing was not specific as pointed out in the detailed comments.

Moreover, the methodology is poorly described and specifically the imbalance of the classes has not been addressed (or is not reported how it was addressed) upon model training, leading to large deviations between results from different cross validation folds and between model sensitivity and specificity. This needs to be improved prior to publication. Since the code has not been made available it is likely impossible to reproduce the reported findings. Overall, I think there could potentially be interesting additional findings if the patient subgroups are reported also individually and a final interpretability analysis of the used model would be presented. There should be a bit more link to the clinical importance of the presented results.

In summary, the manuscript has potential for publication given that the methodology is sound – which was not possible to clearly judge at this stage as important information and results (uncertainty of the evaluation metrics, subgroup results) were not reported.

Detailed comments:

Abstract l.34-37 reads like an introduction and the following methods, results and discussion are mixed. Please clearly separate these sections and ensure neutral, scientific language.

Introduction:  

  • l.73: what ‘other characteristics’ do you refer to here? This is not very specific.
  • l.83/84: How would ML be used to identify the best treatment for an individual tumor? This is not necessarily true, since most ML applications target classification and do not specifically suggest what to do if e.g. the prediction is towards ‘no treatment response’.
  • l.85/85: ‘based on the histogram, textural, and geometric fea-85 tures obtained via radiomic database’ – be more specific. What features for which imaging modality? What is ‘radiomic database’?
  • What do you mean by ‘the predictors were incapable of altering the histopathological methods in the tumor genetic profiling’ 89/90?
  • l.94: ‘the overtreatment-induced adverse effects’ were not previously defined. Although I guess I know what you mean, this is not written very clearly.
  • l.98 and following: This is still introduction and no results should be reported here. Instead clearly state your hypothesis/research question – what is the novelty of this study?

Results:

  • l.120: how was significance tested? Clearly state all statistical methods in the methods section.
  • Table 1: there seems to be a strong mismatch between Sensitivity and specificity – report the prevalence of the positive class.
  • Figure2: How does this ACC compare to the values reported in table 2 which were a lot lower? Was this averaged over multiple cross-validation folds? Using a stronger regularization, shouldn’t it be possible to learn the contribution of features rather than manually selecting them?

  • l.172: Apparently a 5-fold CV was used (please add to the methods, not the results) but no uncertainties are reported. Please include these for all parameters reported in any table. In particular Table 3. It is also unclear what these values are – assumingly the validation subset within each CV fold – since sometimes this dataset is referred to as ‘training’. To prevent confusion, I recommend to use ‘training’ if actual model training was performed, ‘validation’ for the validation subset of the dataset for each CV fold. ‘external test set’ for the external data set.

  • Please also report the prediction scores for patient subgroups by tumor type and grade.

  • l.193 and following: The external validation set does not seem to be a comparable representation of the training set with respect to class imbalance, grade and tumor type (table 4) – it is questionable to just compare acc, specificity etc. for the group as a whole. Please report details for subgroups to understand the underlying reason for the poor sensitivity score.

  • Include a final interpretability analysis of the final model to evaluate the importance of the different features on the actual task.

Discussion:

  • The discussion starts with a repetition of methods – please move this to the methods. (l. 205-211). Followed by a repetition of the results – but what is the key message? What is the novelty?

  • Report uncertainties for the model predictions at all times.

  • The discussion touches on the most important parts in terms of comparison with the literature, however the strongest limitation of the model, i.e. a very poor sensitivity score was hardly discussed at all. It is important when comparing models to also account for differences in the class imbalance, and where possible, to include the prevalence of the positive class.

  • Please try to also interpret your results. There is little actual discussion, but rather only a listing of literature results. Evaluate and discuss the interpretability analysis and subgroup results with respect to the clinical hallmarks of LGG. What do you suggest could be the reason why wavelet features were superior to e.g. intensity-based features?

Methods:

  • l.289-299: Reference table 4 earlier and report values in table 4 also as % rather than absolute numbers. Which class corresponded to which physiological feature? (mentioned only in the discussion). How were splits performed?

  • l.311: It is not clear what you mean – please be more specific. What is ‘the software’ here? How did you validate ‘the exact margin of the tumor’? Were manual contours available?

  • Overall there is far too little information regarding the input data: What was the magnetic field strength and which MR imaging contrasts were used? Were images registered to a reference space/how were the single axial slices selected from the 3D MR image? Were image intensities normalized? Which plane was used and why (coronal, transversal or sagittal)? What was the image resolution and dimension range? Include some details on the relevant scanner and how images were preprocessed (e.g. bias field correction?). Give details on the hyperparameters used for feature extraction – just stating the used package is insufficient to reproduce this work.

  • l.318: What is ‘MR assay’? If this imaging contrast (T1, T2, T2*) were different contrasts combined?

  • l.341: SHAP analysis: It is unclear here which model was used for the SHAP analysis to arrive at the subset of features. From the results section it seems that this was only the XGBboost model, but which data were used in the feature selection process (only the training data should be used here). How did you prevent overfitting (early stopping?) and account for class imbalance?

  • l.355 and following: Give details on the implementation of the different models and make the relevant code available. How many folds were considered for grid search cross validation and which hyperparameters were optimized?

  • l.397: Given the class imbalance, it would be important to also report AUPRC /average precision scores, and to clearly state the prevalence of the dataset. Was some form of class-balancing (e.g. using weighting) performed during training? Given the strong variation in model performance by CV fold, it seems that there are other factors apart from the label driving classification, too. Ensure a balanced split with respect to all important features of the patient cohort(such as grade and tumor type), and not only by label.

  • There is no information on when the external validation set was used and how the models were specifically trained (which hardware, which optimizers, which initialization, learning rater, hyperparameters, stopping criteria). What criteria were used for hyperparameter optimization? How many repeat runs were performed? No information regarding the statistical comparison of different ML models are given. Separate methods section should define these points for all of the ML models used, both for the feature selection optimization, and the initial model training to select XGBoost.

Minor

  1. l.70: ‘the assays’ – replace with ‘the pathology samples’ or ‘tumor biopsies’, an assay is something different.
  2. l.71 ‘contain risk’ – please rephrase.
    There are a number of small language specific problems which exceed the scope of my comments at this stage. Please turn to a native speaker/editing service for support to improve this.

Author Response

We thank the reviewer #1 for deep and thorough review. We have revised our present research paper in the light of reviewer’s useful suggestions and comments. In general, we have revised our manuscript regarding English and phrasing issues. We have also added more details to the methodology of our work to make the framework clearer for readers.

In detailed comments, based on your comments and suggestions, we have clarified and explained some points as following:

  1. Abstract l.34-37 reads like an introduction and the following methods, results and discussion are mixed. Please clearly separate these sections and ensure neutral, scientific language.
  • We modified the phrases and chose more appropriate scientific language. We also separated the methods, results and discussions sections in the abstract.
  1. 73: what ‘other characteristics’ do you refer to here? This is not very specific.
  • “Other characteristics” refers to “genomic biomarkers” and “pathognomonic imaging changes”
  1. 83/84: How would ML be used to identify the best treatment for an individual tumor? This is not necessarily true, since most ML applications target classification and do not specifically suggest what to do if e.g. the prediction is towards ‘no treatment response’.
  • The application of machine learning (ML) to predicting personalized therapy for cancer patients is very promising. Many studies [1-4] demonstrated the predictive capacity of ML in cancer therapy based on big data with information on genomics and pharmacogenomics. Thus, we believe that ML can assist in both diagnosis and individualized treatment for patients.
  1. 85/85: ‘based on the histogram, textural, and geometric fea-85 tures obtained via radiomic database’ – be more specific. What features for which imaging modality? What is ‘radiomic database’?
  • 85: Make clear for “features” and modified “radiomic database” to “radiomics features from medical images”.

What do you mean by ‘the predictors were incapable of altering the histopathological methods in the tumor genetic profiling’ 89/90?

  • 89,90: Modified “were not capable of altering” to “could not replace”.
  1. l.94: ‘the overtreatment-induced adverse effects’ were not previously defined. Although I guess I know what you mean, this is not written very clearly.
  • 94: Modified “overtreatment-induced adverse effects” to “adverse effects induced by invasive diagnostic methods” and make the meaning of the sentence clearer.
  1. l.98 and following: This is still introduction and no results should be reported here. Instead clearly state your hypothesis/research question – what is the novelty of this study?
  • The novelty and difference of our study is that our model can predict the 1p/19q codeletion mutation on Low Grade Gliomas (LGG) patients using only seven radiomic features from medical images. We also excluded the results out of the Introduction section.
  1. 120: how was significance tested? Clearly state all statistical methods in the methods section.
  • 120: Actually, there were no statistical tests applied, the results were observed and compared only. We deleted the word “significantly” to avoid misunderstanding.
  1. Table 1: there seems to be a strong mismatch between Sensitivity and specificity – report the prevalence of the positive class.
  • Table 1: The table illustrated some metrics for the overview of four baseline algorithms. Our purpose is to explain the reason we chose XGBoost to build our baseline model (with higher metric scores) rather than other algorithms.
  1. Figure2: How does this ACC compare to the values reported in table 2 which were a lot lower? Was this averaged over multiple cross-validation folds? Using a stronger regularization, shouldn’t it be possible to learn the contribution of features rather than manually selecting them?
  • The accuracy in Figure 2 is the average over 5-fold cross-validation performance. We have checked again and seen that the accuracy was consistent with the ones in Table 2. Please check again in the manuscript since we aimed to plot accuracy metric here, not sensitivity or AUC. Also, Figure 2 illustrated the performance of XGBoost algorithm with one, two, three… to 37 top radiomic features from the previous step of features selection using XGBoost and SHapley Additive exPlanations (SHAP) analysis. The peak of accuracy was the performance of XGBoost and seven top features.
  1. 172: Apparently a 5-fold CV was used (please add to the methods, not the results) but no uncertainties are reported. Please include these for all parameters reported in any table. In particular Table 3. It is also unclear what these values are – assumingly the validation subset within each CV fold – since sometimes this dataset is referred to as ‘training’. To prevent confusion, I recommend to use ‘training’ if actual model training was performed, ‘validation’ for the validation subset of the dataset for each CV fold. ‘external test set’ for the external data set.
  • We performed 5-fold cross-validation, the values were the AUCs of each fold (validation subset). We complemented the information of uncertainties (AUC) in Table 3.
  • Also, we modified “external validation set” to “external test set” to avoid misunderstandings.
  1. Please also report the prediction scores for patient subgroups by tumor type and grade.
  • We have added the results table as suggested by the reviewer in section 2.5. Please check again in our revised
  1. 193 and following: The external validation set does not seem to be a comparable representation of the training set with respect to class imbalance, grade and tumor type (table 4) – it is questionable to just compare acc, specificity etc. for the group as a whole. Please report details for subgroups to understand the underlying reason for the poor sensitivity score.
  • We have added the results table as suggested by the reviewer in section 2.5. Please check again in our revised manuscript.
  • The data in the external test set was also imbalanced, however, this is necessary to test the performance of our model on unseen and imbalanced data.
  1. Include a final interpretability analysis of the final model to evaluate the importance of the different features on the actual task.
  2. The discussion starts with a repetition of methods – please move this to the methods. (l. 205-211). Followed by a repetition of the results – but what is the key message? What is the novelty?

à We have modified to avoid misunderstandings between sections. The novelty and difference of our study is that our model can predict the 1p/19q codeletion mutation on Low Grade Gliomas (LGG) patients using only seven radiomic features from medical images, compared to other models using more features or not achieving a high accuracy.

  1. The discussion touches on the most important parts in terms of comparison with the literature, however the strongest limitation of the model, i.e. a very poor sensitivity score was hardly discussed at all. It is important when comparing models to also account for differences in the class imbalance, and where possible, to include the prevalence of the positive class.
  • We have added the discussions about the imbalanced results. In the revised manuscript, we also mentioned that we used SMOTE technique to handle imbalance problem.
  1. Please try to also interpret your results. There is little actual discussion, but rather only a listing of literature results. Evaluate and discuss the interpretability analysis and subgroup results with respect to the clinical hallmarks of LGG. What do you suggest could be the reason why wavelet features were superior to e.g. intensity-based features?
  • Wavelet features have advantages in image compression and preprocessing [5-7].
  1. 289-299: Reference table 4 earlier and report values in table 4 also as % rather than absolute numbers. Which class corresponded to which physiological feature? (mentioned only in the discussion). How were splits performed?
  • We have revised the information in Table 4. Please check again in the revised manuscript.
  • We have added the ground-truth labels in the methodology.
  • We used 5-fold cross-validation in evaluating the performance of the data.
  1. 311: It is not clear what you mean – please be more specific. What is ‘the software’ here? How did you validate ‘the exact margin of the tumor’? Were manual contours available?
  • The data we collected was publicly available on TCIA. Also the regions-of-interest was segmented by Akkus et al. [8, 9] using their own semi-automated model. We did not segment these data, therefore, we have re-written these in the manuscript.
  1. Overall there is far too little information regarding the input data: What was the magnetic field strength and which MR imaging contrasts were used? Were images registered to a reference space/how were the single axial slices selected from the 3D MR image? Were image intensities normalized? Which plane was used and why (coronal, transversal or sagittal)? What was the image resolution and dimension range? Include some details on the relevant scanner and how images were preprocessed (e.g. bias field correction?). Give details on the hyperparameters used for feature extraction – just stating the used package is insufficient to reproduce this work.
  • The data we collected was publicly available on TCIA. Also the preprocessing, normalization was done by Akkus et al. [8, 9]. Thus, we used their images in our implementation without changing the original images. We only extracted the radiomics features and used them in our next analyses using machine learning.
  1. 318: What is ‘MR assay’? If this imaging contrast (T1, T2, T2*) were different contrasts combined?
  • We modified “assay” to “records”. It was not a combination of different contrasts.
  1. 341: SHAP analysis: It is unclear here which model was used for the SHAP analysis to arrive at the subset of features. From the results section it seems that this was only the XGBboost model, but which data were used in the feature selection process (only the training data should be used here). How did you prevent overfitting (early stopping?) and account for class imbalance?
  • We used XGBoost algorithm plus SHAP analysis on the training dataset after handling the data imbalanced with SMOTE. We did not implement “early stopping” in this experiment.
  1. 355 and following: Give details on the implementation of the different models and make the relevant code available. How many folds were considered for grid search cross validation and which hyperparameters were optimized?
  • In the revised version, we have included how we implemented different model as well as their hyperparameters. Also, the grid search cross-validation has been done via 5-fold. Finally, the code is available at our GitHub: https://github.com/khanhlee/LGG-radiomics.

  1. 397: Given the class imbalance, it would be important to also report AUPRC /average precision scores, and to clearly state the prevalence of the dataset. Was some form of class-balancing (e.g. using weighting) performed during training? Given the strong variation in model performance by CV fold, it seems that there are other factors apart from the label driving classification, too. Ensure a balanced split with respect to all important features of the patient cohort(such as grade and tumor type), and not only by label.
  • We have added the results in section 2.5. Especially, we have included Precision-Recall curve as well as AUPRC in the experimental results.
  1. There is no information on when the external validation set was used and how the models were specifically trained (which hardware, which optimizers, which initialization, learning rater, hyperparameters, stopping criteria). What criteria were used for hyperparameter optimization? How many repeat runs were performed? No information regarding the statistical comparison of different ML models are given. Separate methods section should define these points for all of the ML models used, both for the feature selection optimization, and the initial model training to select XGBoost.
  • After model implementation (training process and hyperparameter optimization are done), we used that model to predict the external validation set to evaluate the performance. Also, we considered performance of different machine learning models via different measurement metrics. If any model outperformed the others in most metrics, we could consider that model was significant. Therefore, we did not apply any statistical test in the comparison.
  1. l.70: ‘the assays’ – replace with ‘the pathology samples’ or ‘tumor biopsies’, an assay is something different.
  • We have revised and modified the word following your suggestions.
  1. 71 ‘contain risk’ – please rephrase.

We have rephrased the sentence to make it clearer.

References:

  1. Rafique, R., S.M.R. Islam, and J.U. Kazi, Machine learning in the prediction of cancer therapy. Computational and Structural Biotechnology Journal, 2021. 19: p. 4003-4017.
  2. Cuccarini, V., et al., Advanced MRI may complement histological diagnosis of lower grade gliomas and help in predicting survival. 2016. 126(2): p. 279-288.
  3. Huang, C., et al., Machine learning predicts individual cancer patient responses to therapeutic drugs with high accuracy. 2018. 8(1): p. 1-8.
  4. Iqbal, M.J., et al., Clinical applications of artificial intelligence and machine learning in cancer diagnosis: looking into the future. 2021. 21(1): p. 1-11.
  5. Chaddad, A., P. Daniel, and T.J.F.i.o. Niazi, Radiomics evaluation of histological heterogeneity using multiscale textures derived from 3D wavelet transformation of multispectral images. 2018. 8: p. 96.
  6. Dettori, L., L.J.C.i.b. Semler, and medicine, A comparison of wavelet, ridgelet, and curvelet-based texture classification algorithms in computed tomography. 2007. 37(4): p. 486-498.
  7. Weyn, B., et al., Automated breast tumor diagnosis and grading based on wavelet chromatin texture description. 1998. 33(1): p. 32-40.
  8. Akkus, Z., et al., Predicting deletion of chromosomal arms 1p/19q in low-grade gliomas from MR images using machine intelligence. 2017. 30(4): p. 469-476.
  9. Akkus, Z., et al., Semi-automated segmentation of pre-operative low grade gliomas in magnetic resonance imaging. 2015. 15(1): p. 1-10.

If you have more questions or suggestions regarding our work, we would be delighted to answer and explain all of them.

We are looking forward for your reply. Thank you for your time, suggestions and considerations.

Sincerely yours,

The authors of the manuscript.

Reviewer 2 Report

Comment 1 (section 'Simple Summary')

Authors state "however, the current diagnostic gold standard remains highly invasive and time-consuming". Should be useful to specify what the gold standard consists of.

Comment 2 (section 'Abstract')

 - "IDH" acronym is used without having never still introduced

 - "SHAP"  acronym is used without having never still introduced

Comment 3 (section 'Introduction')

From the sentence"Patients of this LGG type..." it is not clear what type of LGG the authors refer to.

Comment 4  (section 'Introduction')

Could authors explain better the sentence "Additionally, the results interpreted from the cytological analysis of LGGs lack critical information about other characteristics of gliomas or other brain cancers"

Comment 5  (section 'Introduction')

With the sentence "More importantly, our proposed 100 model maintained a stable predictive efficacy on the validation set" authors intent 'external validation set'?

Comment 6 (section 2.1. 'Baseline comparison among different machine learning algorithms').

Could authors explain better the sentence  "To evaluate which algorithms worked well on these radiomics features, we executed applying five ML algorithms on the 851-feature training database"

Comment 7 (section 2.1. 'Baseline comparison among different machine learning algorithms').

Authors state  "To evaluate which algorithms worked well on these radiomics features, we executed applying five ML algorithms on the 851-feature training database". Did ML algorithms applied on all 851 features without any skimming (e.g. redundant features, correlated features, without information content features,...)?

Comment 8 (section 2.2. 'Radiomics signature building')

The authors state "In short, this meant that the wavelet-LLL-firstorder-Skewness by itself could precisely classified 48% of the 1p/19q status...". In Figure 1, it is not clear how 48% is obtained.

Comment 9 (section 2.2. 'Radiomics signature building')

The authors state "We intended to determine the optimal number of figures resulting in the most superior accuracy performed by the model". I think tha authors intend "We intended to determine the optimal number of features resulting in the most superior accuracy performed by the model.

Comment 10 (section 2.2. 'Radiomics signature building')

The result shown in figure 2 is obtained by considering and taking the 37 features in the order of the SHAP analysis? Therefore the best accuracy is obtained by considering the first 7 features in the SHAP order?

Comment 11 (section 2.2. 'Radiomics signature building')

Authors state: "Surprisingly, all of the last seven features were in Wavelet form". The last 7 or the first 7?

Comment 12 (section 2.2. 'Radiomics signature building')

To improve readability, I suggest reporting the data in rows 161-171 ("The first-order matrix includes 19 features, by which the distribution of voxel intensities within the ROI mask is described through common statistical metrics. Skewness of the First Order Matrix refers to the measurement of figure distribution’s symmetry around the Mean value (i.e., the average gray level of intensity that lies within the ROI). Depending on the symmetry, this feature could be positive or negative. In terms of describing the shape of distribution, Kurtosis is used to manifest the heaviness of the distribution tail: this feature is positive (i.e., leptokurtic) if the mass of data falls to the tail(s) and not focusing on the Mean value; otherwise, it returns a negative index (i.e., platykurtic). More detailed annotations about the GLCM’s Idn (i.e., Inverse difference normalized), GLDM’s Dependence Non Uniformity Normalized, GLRLM’s Run Variance, and NGTDM’s Busyness could be found in [32].") within a table.

Comment 13 (section 2.2. 'Radiomics signature building')

What means "This could be explained as the lack of samples in our database". Authors intend that the samples are few? If yes, do authors considered a data-augmentation procedure?

Comment 14 (section 2.2. 'Radiomics signature building')

In figure 3 a mean AUC of 0.85±0.08 is illustred, while successively authors state "However, an AUC of 0.88 ± 0.06 was reasonable and agreeable with our values of accuracy, sensitivity, and specificity". It is possible to have further information on 2 reported values.

Comment 15 (section 2.3. 'Comparison to previous studies on 1p/19q status prediction')

In Table 3, dataset 'LGG 159' for Akkus et al. is the same used for the proposed approach? Moreover, I suggest to specifiy (for each work in the comparison) if the dataset is proprietary or freely available.

Comment 16 (sections 2.2. 'Radiomics signature building' and 2.4. 'Performance results on validation cohort')

In these sections 'd/d' and 'n/n'  are used without having explained their meaning (done in  section 3. Discussion).

Comment 17 (section 4.1 'Patient cohort')

Within TCIA 159 LGG databased there are 57 patients with non-deleted 1p/19q condition and 102 co-deleted patients.  This represent an evident scenario of unbalanced classes. Did authors speak about?  Did they take any 'precautions' about it during the training phase?

Comment 18 (section 4.2. 'ROI segmentation')

Could authors explain better the sentences "Our project exploited the previous semi-segmented model by Akkus et al. [23] to segment the lesional region.The T1-weighted and T2-weighted images were subsequently inserted into the software, and the ROI would be automatically shrunk to ultimately cover the exact margin of the tumor".In the first sentence it seems that a semi-automatic model (Akkus et al. [23]) has been used, and that this segmentation is then 'refined' by an undefined 'software'. Rephrase and integrate properly the sentences in order to make them more understandable.

Comment 19 (section 4.3. 'Radiomics feature extraction')

"Zwanenburg et al. and Van Griethuysen et al. [32,36]" -> "Zwanenburg et al. [32] and Van Griethuysen et al. [36]".

Comment 20 (section 4.3. 'Radiomics feature extraction')

Explain how the 3D Slicer software with its seven extensions allows you to extract features, exploiting Pyradiomics.

Comment 21 (section 4.5.2. 'kNN')

There is a reference for "Evelyn Fix and Joseph Hodges introduced an algorithm called ..."?

Comment 22 (section 5. 'Conclusion')

Conclusion are minimal and for this reason must be properly integrated. I suggest to the authors to add some points concerning future development about this work.

Moreover, it might be interesting to discuss regarding automated brain tumor segmentation, the impact and the potential of advanced automatic segmentation approaches using classic Machine Learning [Rundo, L., Militello, C., Russo, G., Vitabile, S., Gilardi, M. C., & Mauri, G. (2018). GTV cut for neuro-radiosurgery treatment planning: an MRI brain cancer seeded image segmentation method based on a cellular automata model. Natural Computing, 17(3), 521-536. DOI: 10.1007/s11047-017-9636-z] or Deep Learning [Pereira, S., Pinto, A., Alves, V., & Silva, C. A. (2016). Brain tumor segmentation using convolutional neural networks in MRI images. IEEE Transactions on Medical Imaging, 35(5), 1240-1251. DOI: 10.1109/TMI.2016.2538465] might be mentioned and discussed. This would be important for radiomic feature reliability.

Comment 23

An English revision of the whole manuscript is mandatory.

Author Response

Dear Editor and Reviewer Two,

We are the authors of the manuscript “Development and validation of an efficient MRI radiomics signature for improving the predictive performance of 1p/19q co-deletion in lower-grade gliomas”.

In general, we have revised our manuscript regarding English and phrasing issues. We have also added more details to the methodology of our work to make the framework clearer for readers.

In detailed comments, based on your comments and suggestions, we have clarified and explained some points as following:

Comment 1 (section 'Simple Summary')

Authors state "however, the current diagnostic gold standard remains highly invasive and time-consuming". Should be useful to specify what the gold standard consists of.

  • We have added the information: “the brain-tissue biopsy or the surgical resection of the tumor”

Comment 2 (section 'Abstract')

 - "IDH" acronym is used without having never still introduced

 - "SHAP"  acronym is used without having never still introduced

  • We have revised and added the information.

Comment 3 (section 'Introduction')

From the sentence"Patients of this LGG type..." it is not clear what type of LGG the authors refer to.

  • We made clear for “this LGG type”. Please check again in the revised version.

Comment 4  (section 'Introduction') Could authors explain better the sentence "Additionally, the results interpreted from the cytological analysis of LGGs lack critical information about other characteristics of gliomas or other brain cancers"

  • The results interpreted from the cytological analysis of LGGs lack critical information about genomic biomarkers or pathognomonic imaging changes of gliomas or other brain cancers.

Comment 5  (section 'Introduction')

With the sentence "More importantly, our proposed 100 model maintained a stable predictive efficacy on the validation set" authors intent 'external validation set'?

  • By “The validation set”, the authors meant “External validation set”. We have modified “the validation set” to “the external test set”.

Comment 6, 7 (section 2.1. 'Baseline comparison among different machine learning algorithms').

Could authors explain better the sentence  "To evaluate which algorithms worked well on these radiomics features, we executed applying five ML algorithms on the 851-feature training database"

Authors state  "To evaluate which algorithms worked well on these radiomics features, we executed applying five ML algorithms on the 851-feature training database". Did ML algorithms applied on all 851 features without any skimming (e.g. redundant features, correlated features, without information content features,...)?

  • In the feature selection process, we used SHAP analysis to rank features based on their importance. However, it is necessary to have a baseline model to combine with SHAP so that in can function in the selection. For this purpose, we needed to apply the algorithms on the preliminary training dataset with 851 features and 159 patients to find the most appropriate baseline algorithm.

Comment 8 (section 2.2. 'Radiomics signature building')

The authors state "In short, this meant that the wavelet-LLL-firstorder-Skewness by itself could precisely classified 48% of the 1p/19q status...". In Figure 1, it is not clear how 48% is obtained.

  • In Figure 1: The horizontal axis showed how much the features contribute to the output predictor variable via SHAP values. Features were ranked based on their contribution to the output predictor; the higher the SHAP value, the higher the rank of the corresponding feature. The vertical axis with the color ranged from blue to red represented the value of a feature from low to high, respectively. The density of the colored dots in any SHAP value represented the strength or weakness of the feature in that SHAP value range. The original-firstorder-Skewness feature had the greatest impact on the predictability of the model. SHAP value of original-firstorder-Skewness was 0.48 (absolute value of -0.4 plus +0.08). The density of red and purple points concentrated in the range from 0 to +0.08 indicated that the predictive value of this feature was highest as its corresponding SHAP value was from 0 to +0.08. On the other hand, the feature wavelet-HHH-firstorder-Median ranked last in the graph, with the lowest SHAP value 0.06 (absolute value -0.05 plus +0.01), although it was still a relatively important feature (density thick red and purple dots).

Comment 9 (section 2.2. 'Radiomics signature building')

The authors state "We intended to determine the optimal number of figures resulting in the most superior accuracy performed by the model". I think tha authors intend "We intended to determine the optimal number of features resulting in the most superior accuracy performed by the model.

  • We modified “figures” to “features”.

Comment 10 (section 2.2. 'Radiomics signature building')

The result shown in figure 2 is obtained by considering and taking the 37 features in the order of the SHAP analysis? Therefore the best accuracy is obtained by considering the first 7 features in the SHAP order?

  • Indeed, we tried inputting each feature according to their rank (obtained after implementing SHAP analysis) one-by-one to test which subset of features would be the optimal ones, and they were the seven first features.

Comment 11 (section 2.2. 'Radiomics signature building')

Authors state: "Surprisingly, all of the last seven features were in Wavelet form". The last 7 or the first 7?

  • It was “the first seven features”, or “top seven features”, we have modified “last” to “first”. And there were actually one original feature and six wavelet features. We have revised and modified.

Comment 13 (section 2.2. 'Radiomics signature building')

What means "This could be explained as the lack of samples in our database". Authors intend that the samples are few? If yes, do authors considered a data-augmentation procedure?

  • We did not implement data-augmentation procedure, since this is the data for external validation. In the training process, we applied SMOTE to address the data imbalance.

Comment 14 (section 2.2. 'Radiomics signature building')

In figure 3 a mean AUC of 0.85±0.08 is illustred, while successively authors state "However, an AUC of 0.88 ± 0.06 was reasonable and agreeable with our values of accuracy, sensitivity, and specificity". It is possible to have further information on 2 reported values.

  • We have revised and modified. The right result was 0.85±0.06.

Comment 15 (section 2.3. 'Comparison to previous studies on 1p/19q status prediction')

In Table 3, dataset 'LGG 159' for Akkus et al. is the same used for the proposed approach? Moreover, I suggest to specifiy (for each work in the comparison) if the dataset is proprietary or freely available.

  • We have added the references. The database for our present study was the same database LGG 159 from Akkus et al. on TCIA [1].

Comment 16 (sections 2.2. 'Radiomics signature building' and 2.4. 'Performance results on validation cohort')

In these sections 'd/d' and 'n/n' are used without having explained their meaning (done in  section 3. Discussion).

  • We have added the information of ground-truth labels in Materials and Methods section.

Comment 17 (section 4.1 'Patient cohort')

Within TCIA 159 LGG databased there are 57 patients with non-deleted 1p/19q condition and 102 co-deleted patients.  This represent an evident scenario of unbalanced classes. Did authors speak about?  Did they take any 'precautions' about it during the training phase?

  • We implemented SMOTE to address the data imbalance and added the information in methodology.

Comment 18 (section 4.2. 'ROI segmentation')

Could authors explain better the sentences "Our project exploited the previous semi-segmented model by Akkus et al. [23] to segment the lesional region.The T1-weighted and T2-weighted images were subsequently inserted into the software, and the ROI would be automatically shrunk to ultimately cover the exact margin of the tumor".In the first sentence it seems that a semi-automatic model (Akkus et al. [23]) has been used, and that this segmentation is then 'refined' by an undefined 'software'. Rephrase and integrate properly the sentences in order to make them more understandable.

  • The data we collected was publicly available on TCIA. Also the regions-of-interest was segmented by Akkus et al. [1, 2] using their own semi-automated model. We did not segment these data; therefore, we have re-written these in the manuscript.

Comment 20 (section 4.3. 'Radiomics feature extraction')

Explain how the 3D Slicer software with its seven extensions allows you to extract features, exploiting Pyradiomics.

  • We have added the information.

Comment 21 (section 4.5.2. 'kNN')

There is a reference for "Evelyn Fix and Joseph Hodges introduced an algorithm called ..."?

  • We have added the reference.

Comment 22 (section 5. 'Conclusion')

Conclusion are minimal and for this reason must be properly integrated. I suggest to the authors to add some points concerning future development about this work.

  • We have added our future plan.

Moreover, it might be interesting to discuss regarding automated brain tumor segmentation, the impact and the potential of advanced automatic segmentation approaches using classic Machine Learning [Rundo, L., Militello, C., Russo, G., Vitabile, S., Gilardi, M. C., & Mauri, G. (2018). GTV cut for neuro-radiosurgery treatment planning: an MRI brain cancer seeded image segmentation method based on a cellular automata model. Natural Computing, 17(3), 521-536. DOI: 10.1007/s11047-017-9636-z] or Deep Learning [Pereira, S., Pinto, A., Alves, V., & Silva, C. A. (2016). Brain tumor segmentation using convolutional neural networks in MRI images. IEEE Transactions on Medical Imaging, 35(5), 1240-1251. DOI: 10.1109/TMI.2016.2538465] might be mentioned and discussed. This would be important for radiomic feature reliability.

  • We have discussed about the semi-automated and auto-segmentation procedures developed by the authors in the Discussions section.

Comment 23

An English revision of the whole manuscript is mandatory.

  • We have gone through and revised the manuscript in terms of English writing errors and typos. The reviewers could check again in the whole manuscript. We hope that the writing of manuscript is improved and can meet the quality requirement of journal for publication.

References:

  1. Akkus, Z., et al., Predicting deletion of chromosomal arms 1p/19q in low-grade gliomas from MR images using machine intelligence. 2017. 30(4): p. 469-476.
  2. Akkus, Z., et al., Semi-automated segmentation of pre-operative low grade gliomas in magnetic resonance imaging. 2015. 15(1): p. 1-10.

 If you have more questions or suggestions regarding our work, we would be delighted to answer and explain all of them.

We are looking forward for your reply. Thank you for your time, suggestions and considerations.

Sincerely yours,

The authors of the manuscript.

Reviewer 3 Report

Title: "Development and validation of an efficient MRI radiomics signature for improving the predictive performance of 1p/19q co-deletion in lower-grade gliomas"

In this study, the authors proposed an explainable model using XGBoost algorithm, based on only seven radiomic features, to efficiently predict the 1p/19q codeletion status in a binary classification task. The authors used as training dataset the public TCIA LGG 1p/19q codeletion database, which contained 159 patients. They used XGBoost, in association with SHAP analysis to filter out the seven optimal features to assemble a complete model. Their final model achieved an accuracy of 86.8% and 82.8% on the training set and external validation set, respectively. The authors claim that with solely seven wavelet radiomic features, our XGBoost-based model is able to identify the 1p/19q codeletion status in LGG-diagnosed patients and address the drawbacks of the invasive gold-standard tests, thereby ameliorate long-term management for patients diagnosed with LGG.

General comment:

This work seems to have an interesting aim. Indeed, since patients of a LGG type significantly had a greater amount of survival time and are more sensitive to therapeutics in terms of chemotherapy, and radiotherapy with respect to those with 1p/19q non-deleted tumors, an early diagnosis of LGGs is a relevant scientific aim, as this can facilitate short-term and long-term management. In addition, since the current gold-standard clinic procedure to identify1p/19q codeletion is the examination of the tumor’s histopathological sample, this work could provide further information in a non invasive way. However, some issues should be reworked to further enhance the quality and the impact of this work. In particular, some parts should be better explained to the interested readers also showing some examples on chosen images. In addition, also the “Discussion” and “Conclusion” sections could be enhanced to improve the impact of this work. The language is also suboptimal in some parts and should be revised to help the interested readers to better follow the logic flow of the main text of this work.

Some detailed comments:

Section “2.Results”

*) This section, which is the main section of the work should be enlarged to better explain all the relevant results. In addition, only the original results achieved in this work should presented without comments. All the comments, should be transferred to the “Discussion” section.

Figure 1. Top 20 features extracted using SHAP analysis. The wavelet-LLL-firstorder-Skewness was 144 the most appropriate feature contributing to the prediction of the XGB-based model.

*) This figure is not clear, please improve the caption to explain in a detailed way all the main “features” of this figure.

Table 2. 7 radiomics features using in our final model

*) This table is not clear, please provide a more explaining caption. Please provide also the physical meaning of each feature (e.g., what are Busyness and Idn ?)

Figure 2. Performance results when using different numbers of top-rank features in predicting

1p/19 status using XGBoost algorithm. The optimal cut-off point belongs to seven features (accuracy=86.8%).

*) This figure is only partially clear since the different combinations of features should be specified in all details.

Figure 3. ROC curve analysis in predicting 1p/19q status using XGBoost classifier on top seven
radiomics features.

*) This figure is definitely not clear. A more detailed caption should be inserted. Please explain better all the information whitin the Legend.

Lines: “2.4. Performance results on validation cohort 193

We applied our optimal to the validation data and the results showed that our model 194

classified sharply 82.80% of 1p/19q codeleted LGG. However, due to the imbalance of the 195

validation cohort (i.e., 13 d/d LGG vs. 52 n/n LGG), the sensitivity and specificity observed 196

wide variations. While the specificity experienced 94.10%, only 33.30% was for the sensi- 197

tivity. Despite the contrast in the mentioned metrics, the result was plausible. Moreover, 198

the level of 82.8% indicated that there was no overfitting, compare with the predictive rate 199

of 86.8% obtained from the performance of XGBoost-7 on the training set.

More importantly, our framework yielded that, with only 7 features, we could be 201

capable of creating a model which reasonably identified the status of 1p/19q among the 202

LGG cohort.”

*) The authors should better explain why “due to the imbalance of the 195

validation cohort (i.e., 13 d/d LGG vs. 52 n/n LGG), the sensitivity and specificity observed 196

wide variations. While the specificity experienced 94.10%, only 33.30% was for the sensi- 197

tivity. Despite the contrast in the mentioned metrics, the result was plausible.”

Materials and methods

Lines: “4. Materials and Methods 266

Our workflow (which was exhibited in Figure 4), from data extraction to the external 267

validation, comprised three main steps: (I) Data extraction from two public databases 268

from former articles [10,31], one for training and the other one for validation; (II) Radiomic 269

features extraction and the feature refinement using SCC, XGBoost, and SHAP analysis; 270

(III) Applying ML algorithm on refined features to predict the 1p/19q codeletion status of 271

LGG patients on training and validation set.”

and

Figure 4. The workflow of our study. (I) Data extraction from two public database from former 274

articles, one for training and the other one for validation; (II) Radiomic features extraction and the feature refinement using SCC, XGBoost, and SHAP analysis; (III) Applying machine learning algo-rithm XGBoost classifier on refined features to predict the 1p/19q codeletion status of LGG pa- tients on training and validation set.

*) The authors should better explain what is the main procedure which has been followed to extract the main features from the images database. In particular, the current explanation seems to be lacking and interested readers can not totally understand. What are the main procedures to extract from TCIA ,851 radiomic features ? What are the relationships between tumour distribution, intensity, shape and wavelet filter ? How many features of the 851 belonged to each class. Similarly, also the data-mining process should be made more clear (at least in Figure4, but also within the main text of the manuscript). It is not clear how, from SHAP analysis, 37 features have been extracted and optimized in 7 main representative features. What is the exact meaning of “Best performance features” ? Finally, also the analysis should be better explained to the interested readers. Indeed, the third part of the panel in Figure 4 is not totally clear, both with respect to the Xboost (please describe in more details the 5 fold cross validation), and for the external validation (TCIA dataset 65?). Also the plot, as already said, is not too clear. Please rework this part of the main text accordingly.

Lines: “4.1. Patient cohort 279

Our training cohort has been derived from the previous study [10] that included a 280

total of 159 LGG in the T1 and T2 sequence. The original database was published on TCIA 281

Public Access on Sep 30, 2017. Since it is a public database for researching, these data are 282

approved by the Institutional Review Board (IRB) for ethical issues. In this study, we 283

included the version-2 database, which was updated on June 26, 2020, and contained 159 284

patients with confirmed preoperative diagnosis, histopathological result of LGG, and 285

1p/19q status. 286

The downloading process required the installation of the NBIA Data Retriever (ver- 287

sion 3.6, released in April 2020). There were 57 patients with non-deleted 1p/19q condition 288

and 102 co-deleted patients. Among 102 co-deleted patients, there were 66 grade-II and 289

36 grade-III patients, compared with 38 and 19 patients in the respective grades of the 290

non-deleted group. In terms of the tumor types, i.e., Astrocytoma, oligoastrocytoma, and 291

oligodendroglioma, the number of patients diagnosed with oligoastrocytoma accounted 292

for the highest proportion of 61% regardless of the 1p/19q mutated condition, while peo- 293

ple suffering from astrocytoma made up only 10.7% total patients. 294

The external validation cohort was attained from previous research by Bakas et al. 295

[31], which consisted of 65 LGG patients, with confirmed genetic examinations of 1p/19q 296

codeletion status. In detail, six of thirteen 1p/19q-co-deleted and 22 of 52 1p/19q-non-de- 297

leted patients were WHO-Grade II LGGs; and all of the thirteen 1p/19q-co-deleted LGGs 298

were oligodendrogliomas. 299

The MR records of patients in the training and external validation cohort were in 300

axial plane only and T1- or T2-weighted. Detailed information of patients was shown in 301”

*) it seems that the authors used a public database. So the information “Since it is a public database for researching, these data are approved by the Institutional Review Board (IRB) for ethical issues.” is really needed ? Please explain better.

Lines: “4.2. ROI segmentation 305
In the field of MR radiomics, the identification of ROI, which is conceptualized as an 306
area where the computation of radiomics features occurs [36], is crucial for deeper exam- 307
inations of MR assays. Following the mentioned concept, ROI segmentation could be con- 308
ducted via manually segmented or (semi-)auto segmented based on the development of 309
machinery intelligence algorithms [10,23,36]. Our project exploited the previous semi-seg- 310
mented model by Akkus et al. [23] to segment the lesional region. For the beginning, we 311
drew a line to cover the ROI, some lesion-free areas might be involved. The T1-weighted 312
and T2-weighted images were subsequently inserted into the software, and the ROI 313
would be automatically shrunk to ultimately cover the exact margin of the tumor.”

*) The authors should better explain the procedure used to extract the ROI. What is the exact meaning of the words “via manually segmented or (semi-)auto segmented” may the authors provide an example with a chosen representative image ?

Lines: “4.3. Radiomics feature extraction 315
We took advantage of 3D Slicer software (version 4.10.2; released on Oct 10, 2012; last 316
updated on May 17, 2019), which was used for medical image extraction and visualiza- 317
tion, to extract the features from each single MR assay, via seven extensions (e.g., DCMQI, 318
PETDICOMExtension, QuantitativeReporting, SlicerDevelopmentToolbox, SlicerRadiomics, 319
SlicerRT), and the Pyradiomics [32] module in Python (version 3.8). Initially, the MR rec- 320
ords were orderly imported to the software for extraction. 851 features derived from each 321
record were afterward stored in .tsv-formatted files. More specifically, the features were 322
classified into 9 categories, e.g., original, wavelet HHH, wavelet HHL, wavelet HLH, 323
wavelet HLL, wavelet LHH, wavelet LHL, wavelet LLH, and wavelet LLL. Each category 324
comprised of six sub-categories, namely first-order, Gray Level Co-occurrence Matrix 325
(GLCM), Gray Level Size Zone (GLSZM), Gray Level Run Length Matrix (GLRLM), 326

Neighbouring Gray Tone Difference Matrix (NGTDM), Gray Level Dependence Matrix 327
(GLDM); except the original radiomics category with one sub-categorize (Shape) more 328
than the others. The information about the radiomics classes was concretely described by 329
Zwanenburg et al. and Van Griethuysen et al. [32,36].

*) The authors may better explain these lines for interested readers, and they should provide also an example where all these passages are better described.

Sections:

4.4. Feature selection

4.5. Machine learning implementation

4.5.1. LR

4.5.2. kNN

4.5.3. RF

4.5.4. AdaBoost

4.5.5. XGBoost

4.6. Statistical analysis

*) In this paragraphs the authors should split the description of methods from all the comments related to methods. All comments should be inserted within the “Discussion” section (if novel and related to this work) or within the “Introduction” section if related to already known methods. Please provide also some example and a more detailed description of the used software (also all the needed information to identify the commercial or the free open source software)

Discussion

Conclusions

*) The authors may improve the “Discussion” section with quantitative comparisons to the current state of the art to show the value of their work. Similarly, the “Conclusion” section should be improve to resume in an effective way the value of their scientific contribution.

Author Response

Dear Editor and Reviewer Three,

We are the authors of the manuscript “Development and validation of an efficient MRI radiomics signature for improving the predictive performance of 1p/19q co-deletion in lower-grade gliomas”.

In general, we have revised our manuscript regarding English and phrasing issues. We have also added more details to the methodology of our work to make the framework clearer for readers.

In detailed comments, based on your comments and suggestions, we have clarified and explained some points as following: 

Section “2.Results”

*) This section, which is the main section of the work should be enlarged to better explain all the relevant results. In addition, only the original results achieved in this work should presented without comments. All the comments, should be transferred to the “Discussion” section.

à We have modified and moved the comments to the Discussions section.

Figure 1. Top 20 features extracted using SHAP analysis. The wavelet-LLL-firstorder-Skewness was 144 the most appropriate feature contributing to the prediction of the XGB-based model.

*) This figure is not clear, please improve the caption to explain in a detailed way all the main “features” of this figure.

à We have modified and added the details.

Figure 2. Performance results when using different numbers of top-rank features in predicting

1p/19 status using XGBoost algorithm. The optimal cut-off point belongs to seven features (accuracy=86.8%).

*) This figure is only partially clear since the different combinations of features should be specified in all details.

 à The figure illustrated the accuracy scores in each combination between XGBoost and the features.

Figure 3. ROC curve analysis in predicting 1p/19q status using XGBoost classifier on top seven
radiomics features.

*) This figure is definitely not clear. A more detailed caption should be inserted. Please explain better all the information whitin the Legend.

à The figures illustrated each fold’s AUC. We have added the information.

Lines: “2.4. Performance results on validation cohort 193

We applied our optimal to the validation data and the results showed that our model 194

classified sharply 82.80% of 1p/19q codeleted LGG. However, due to the imbalance of the 195

validation cohort (i.e., 13 d/d LGG vs. 52 n/n LGG), the sensitivity and specificity observed 196

wide variations. While the specificity experienced 94.10%, only 33.30% was for the sensi- 197

tivity. Despite the contrast in the mentioned metrics, the result was plausible. Moreover, 198

the level of 82.8% indicated that there was no overfitting, compare with the predictive rate 199

of 86.8% obtained from the performance of XGBoost-7 on the training set.

More importantly, our framework yielded that, with only 7 features, we could be 201

capable of creating a model which reasonably identified the status of 1p/19q among the 202

LGG cohort.”

*) The authors should better explain why “due to the imbalance of the 195

validation cohort (i.e., 13 d/d LGG vs. 52 n/n LGG), the sensitivity and specificity observed 196

wide variations. While the specificity experienced 94.10%, only 33.30% was for the sensi- 197

tivity. Despite the contrast in the mentioned metrics, the result was plausible.”

 à  Our purpose is to evaluate the performance of our model on unseen data. The results on the external test set containing imbalanced data was reasonable, this indicated that the performance of our model was stable.

Materials and methods

Lines: “4. Materials and Methods 266

Our workflow (which was exhibited in Figure 4), from data extraction to the external 267

validation, comprised three main steps: (I) Data extraction from two public databases 268

from former articles [10,31], one for training and the other one for validation; (II) Radiomic 269

features extraction and the feature refinement using SCC, XGBoost, and SHAP analysis; 270

(III) Applying ML algorithm on refined features to predict the 1p/19q codeletion status of 271

LGG patients on training and validation set.”

and

Figure 4. The workflow of our study. (I) Data extraction from two public database from former 274

articles, one for training and the other one for validation; (II) Radiomic features extraction and the feature refinement using SCC, XGBoost, and SHAP analysis; (III) Applying machine learning algo-rithm XGBoost classifier on refined features to predict the 1p/19q codeletion status of LGG pa- tients on training and validation set.

*) The authors should better explain what is the main procedure which has been followed to extract the main features from the images database. In particular, the current explanation seems to be lacking and interested readers can not totally understand. What are the main procedures to extract from TCIA ,851 radiomic features ? What are the relationships between tumour distribution, intensity, shape and wavelet filter ? How many features of the 851 belonged to each class. Similarly, also the data-mining process should be made more clear (at least in Figure4, but also within the main text of the manuscript). It is not clear how, from SHAP analysis, 37 features have been extracted and optimized in 7 main representative features. What is the exact meaning of “Best performance features” ? Finally, also the analysis should be better explained to the interested readers. Indeed, the third part of the panel in Figure 4 is not totally clear, both with respect to the Xboost (please describe in more details the 5 fold cross validation), and for the external validation (TCIA dataset 65?). Also the plot, as already said, is not too clear. Please rework this part of the main text accordingly.

à We have described the procedure of feature extraction.

à Tumour distribution, intensity, shape and wavelet: We classified the features into 4 categories based on a previous study of Aerts et al. [1]

à From SHAP analysis, 37 features have been extracted and optimized in 7 main representative features: To ensemble the optimal model with the most significant features, we sequentially added the features in order of importance from the highest to the lowest and recorded the accuracy scores corresponding to the number of features to the XGBoost model. The combination of seven top features and XGBoost yielded the highest performance.

à “Best performance features” means the most important features that contribute to the prediction of the model.

Lines: “4.1. Patient cohort 279

Our training cohort has been derived from the previous study [10] that included a 280

total of 159 LGG in the T1 and T2 sequence. The original database was published on TCIA 281

Public Access on Sep 30, 2017. Since it is a public database for researching, these data are 282

approved by the Institutional Review Board (IRB) for ethical issues. In this study, we 283

included the version-2 database, which was updated on June 26, 2020, and contained 159 284

patients with confirmed preoperative diagnosis, histopathological result of LGG, and 285

1p/19q status. 286

The downloading process required the installation of the NBIA Data Retriever (ver- 287

sion 3.6, released in April 2020). There were 57 patients with non-deleted 1p/19q condition 288

and 102 co-deleted patients. Among 102 co-deleted patients, there were 66 grade-II and 289

36 grade-III patients, compared with 38 and 19 patients in the respective grades of the 290

non-deleted group. In terms of the tumor types, i.e., Astrocytoma, oligoastrocytoma, and 291

oligodendroglioma, the number of patients diagnosed with oligoastrocytoma accounted 292

for the highest proportion of 61% regardless of the 1p/19q mutated condition, while peo- 293

ple suffering from astrocytoma made up only 10.7% total patients. 294

The external validation cohort was attained from previous research by Bakas et al. 295

[31], which consisted of 65 LGG patients, with confirmed genetic examinations of 1p/19q 296

codeletion status. In detail, six of thirteen 1p/19q-co-deleted and 22 of 52 1p/19q-non-de- 297

leted patients were WHO-Grade II LGGs; and all of the thirteen 1p/19q-co-deleted LGGs 298

were oligodendrogliomas. 299

The MR records of patients in the training and external validation cohort were in 300

axial plane only and T1- or T2-weighted. Detailed information of patients was shown in 301”

*) it seems that the authors used a public database. So the information “Since it is a public database for researching, these data are approved by the Institutional Review Board (IRB) for ethical issues.” is really needed ? Please explain better.

à We have removed the sentence to avoid misunderstandings.

Lines: “4.2. ROI segmentation 305
In the field of MR radiomics, the identification of ROI, which is conceptualized as an 306
area where the computation of radiomics features occurs [36], is crucial for deeper exam- 307
inations of MR assays. Following the mentioned concept, ROI segmentation could be con- 308
ducted via manually segmented or (semi-)auto segmented based on the development of 309
machinery intelligence algorithms [10,23,36]. Our project exploited the previous semi-seg- 310
mented model by Akkus et al. [23] to segment the lesional region. For the beginning, we 311
drew a line to cover the ROI, some lesion-free areas might be involved. The T1-weighted 312
and T2-weighted images were subsequently inserted into the software, and the ROI 313
would be automatically shrunk to ultimately cover the exact margin of the tumor.”

*) The authors should better explain the procedure used to extract the ROI. What is the exact meaning of the words “via manually segmented or (semi-)auto segmented” may the authors provide an example with a chosen representative image ?

à The data we collected was publicly available on TCIA. Also the regions-of-interest was segmented by Akkus et al. [2, 3] using their own semi-automated model. We did not segment these data; therefore, we have re-written these in the manuscript.

Sections:

4.4. Feature selection

4.5. Machine learning implementation

4.5.1. LR

4.5.2. kNN

4.5.3. RF

4.5.4. AdaBoost

4.5.5. XGBoost

4.6. Statistical analysis

*) In this paragraphs the authors should split the description of methods from all the comments related to methods. All comments should be inserted within the “Discussion” section (if novel and related to this work) or within the “Introduction” section if related to already known methods. Please provide also some example and a more detailed description of the used software (also all the needed information to identify the commercial or the free open source software)

à We have moved the comments to the Discussion section.

Discussion

Conclusions

*) The authors may improve the “Discussion” section with quantitative comparisons to the current state of the art to show the value of their work. Similarly, the “Conclusion” section should be improve to resume in an effective way the value of their scientific contribution.

à We have added the information in the Discussion and Conclusion Section.

References:

  1. Aerts, H.J., et al., Decoding tumour phenotype by noninvasive imaging using a quantitative radiomics approach. 2014. 5(1): p. 1-9.
  2. Akkus, Z., et al., Predicting deletion of chromosomal arms 1p/19q in low-grade gliomas from MR images using machine intelligence. 2017. 30(4): p. 469-476.
  3. Akkus, Z., et al., Semi-automated segmentation of pre-operative low grade gliomas in magnetic resonance imaging. 2015. 15(1): p. 1-10.

If you have more questions or suggestions regarding our work, we would be delighted to answer and explain all of them.

We are looking forward for your reply. Thank you for your time, suggestions and considerations.

Sincerely yours,

The authors of the manuscript.

Round 2

Reviewer 1 Report

The authors have not addressed all recommended changes as outlined below and also have not highlighted any changes in the manuscript which makes it much more elaborate to review the changes that have been included. The revised version of the manuscript, is however, much improved in terms of readability and presentation of the results. I still am however, not convinced by the way the model performance is evaluated (APS should always be included, with relevant uncertainties over CV folds), how patients were split for model taining (this needs to account for tumour subtypes), and that class imbalance seems to have only been accounted for in the final selected model. I also strongly would recommend a direct comparison with the current SOTA model if possible. If these points will be addressed, I think the study may be of interest to the community.

Abstract:

  • The abstract is much improved.

Introduction:

  • 95 and following – please remove all results from the introduction, also qualitative results. Instead state your hypothesis more clearly, why should your approach be superior to the current SOTA?

Result:

  • SMOTE needs to be applied also for model selection. It does not make sense to change the model following it’s identification.
  • Table 1: Still not uncertainties are reported, no AUPRC either. Please add. In the cation state if these are mean values over the 5-fold CV. In particular given the improved specificity, the RF may actually be better in terms of AUPRC than the XGBoost?
  • All performance results should be reported in terms of AUPRC and AUC, too. Add these to figure 2 and also include uncertainty bands in this figure for all of the performance metrics.
  • Figure 3: Include shaded area for the Precision recall curve, too and state uncertainties in the mean scores in the legend. Also add the ‘by chance’ red dashed line in figure 3B.
  • Table 4: Report AUPRC and include uncertainties for the CV. There seems to be a great mismatch in terms of sensitivity between the training and external test set for the different tumour subgroups. This needs to be addressed.
  • Add comparisons of the other models (LR, RF,kNN, AdaBoost) with the selected 37 features, too.
  • A comparison with the current SOTA is missing (this seems to be a deep learning model).
  • If your conclusion should be that your model generalizes better to external data, please show the performance for the combined dataset, too.

Discussion:
- Still no discussion of the tumour subtypes has been included. Please add this.

  • Based on Table 3: there seem to be deep learning models which greatly outperform the presented approach. Please include this in the discussion and if the relevant code is available, report results on your data sets for this SOTA model.
  • It is clear at only 7 wavelet features were included, but why would fewer features be better in a clinical context? Given the increased complexity of generating these features, clinicians might actually prefer models with more, but less complex and more interpretable features. Please address. As such, unfortunately there is very limited improvement of the current study with respect to what has previously been reported.

Methods:

  • Please show table 5 in percent! Here it is clear that the split into training/validation cohorts is not balanced with respect to the subgroups of tumours which performed very differently. Please use a stratified split to ensure balanced distributions in all subcategories in the training/validation set.
  • In section 4.1. also introduce the external test cohort in the same way as the training data and include it also in Table 5.
  • Mention that you performed 5-fold CV.
  • Include brief details on how the relevant models were implemented (e.g. python tensorflow/sklearn etc.) and how hyperparameters were optimized (e.g. GridSearchCV RandomizedSearchCV etc. )
  • Include APS or AUPRC for all models.
  • Include an interpretability analysis of the model using a subset of features.

Typo: l. 69’pathognomonic’

  1. 91: thnce
  2. 300: ‘features extraction’

l.421: new paragraph needed

Reviewer 2 Report

Compared to the original version, this updated version of the manuscript has significantly improved in terms of accuracy, completeness and readability. Comments raised in previous round have all been properly addressed.

Author Response

Thanks for your positive comments.

Reviewer 3 Report

Title: "Development and validation of an efficient MRI radiomics signature for improving the predictive performance of 1p/19q co-deletion in lower-grade gliomas"

In this study, the authors proposed an explainable model using XGBoost algorithm, based on only seven radiomic features, to efficiently predict the 1p/19q codeletion status in a binary classification task. The authors used as training dataset the public TCIA LGG 1p/19q codeletion database, which contained 159 patients. They used XGBoost, in association with SHAP analysis to filter out the seven optimal features to assemble a complete model. Their final model achieved an accuracy of 86.8% and 82.8% on the training set and external validation set, respectively. The authors claim that with solely seven wavelet radiomic features, our XGBoost-based model is able to identify the 1p/19q codeletion status in LGG-diagnosed patients and address the drawbacks of the invasive gold-standard tests, thereby ameliorate long-term management for patients diagnosed with LGG.

General comment: The authors revised this work according to the comments of this reviewer. Only few minor issues

Figure 3. Predicting 1p/19q status using XGBoost classifier on top seven radiomics features. (A) 184
ROC curve analysis, (B) PRC curve analysis.

*) Please, improve this caption

*) Also Tab 4 needs to be better commented within the caption

Round 3

Reviewer 1 Report

The authors have addressed all my previous comments and I think the manuscript is ready for publication, given two minor last changes:
1. My only remaining criticism is with respect to the final discussion of the contribution which I think would be good to rephrase: 

l. 244 "...clinicians may actually use our model to predict 1p/19q mutation 245 status with less time and more accurate than other studies." -  please indicate why less time would be needed for your prediction and why it is more accurate (it does not seem to beat the SOTA)? Given that extensive preprocessing is required to generate features it is questionable if this would really be easier to apply in a clinical setting, please discuss. Given the specific run time (which is also not given for any of the other models) it seems questionable whether the evaluation of a CNN would be a disadvantage over the presented method neither in terms of runtime, nor accuracy. It would be better to clearly state the limitations of the approach in the discussion rather than trying to sell it as the next best model. 

2. 
Intro l. 90: ‘to tackle the limitations of most models’ – it is unclear which limitations of other models a are specifically addressed here, please be more clear.

Author Response

The reviewer is appreciated for fruitful comments. According to the comments, we address them as follows:

1. We have added more discussions to show the advantages of our model in terms of running time compared to the previous deep learning model as follows:

"Shortening the time to detect the type of 1p/19q mutation in patients with LGG is important in terms of the treatment and prognosis of LGG. In clinical practice, it is necessary to shorten the runtime of the model, while ensuring high accuracy in predictions. Our model achieved high performance and stability based on seven optimal features with ten-second runtime for MRI images with tumor segmentation. Problems can arise during the extraction of radiomics features from tumor images. In this study, the time from extracting radiomics features until the prediction was made was 2.8 +- 0.3 minutes, achieving AUC 0.85+-0.06. The deep learning model proposed by Yogananda et al., with a runtime of three minutes (from tumor segmentation to results), achieved the AUC of 0.95 +- 0.01 in predictions. Despite the similar runtime, our model had a lower predictive performance compared to the model proposed by Yogananda et al. In the future, we will focus on improving the model's performance while maintaining or further shortening the total time of the process from image input to output."

Moreover, we have also mentioned that it is one of the limitations of our model in the 'limitation' section as follows:

"However, some limitations still need to be addressed. First, the modest datasets used for training and validating tasks conflicted with the real-world situations, as a large number of LGG patients with known 1p/19q codeletion were not included. For the next steps, data of LGG patients from medical centers would be included to our trained model for evaluation. Second, the binary classification of LGG patients was insufficient, since this study lacked the crucial prognostic marker IDH 1 and 2 mutation [4, 5]. In the future, we would conduct the multiclassification to contribute to preoperative treatment plans. Third, only MR images in axial plane were considered. This unintendedly overlooked some potential characteristics of the tumor. For the next project, the authors would try to propose a model that is compatible with all MRI planes; thence comprehensively predict the 3D shape of the tumor. Fourth, also a constraint of most studies, only retrospective database was considered; hence, the application on prospective cohorts would gauge the robustness of our model more completely. Finally, despite the stability and promising results of XGBoost algorithm towards the unbalanced database [45, 46], and the given result on the training and external validation cohort, the above citations implied that the combination of the present model and deep learning algorithms may significantly enhance the outcomes in the future."

2. We have included the limitations tackling part in the introduction also to address the comments from the reviewer:

"We proposed an eXtreme Gradient Boosting (XGBoost) model to tackle the limitations of most models in terms of prolonged runtime, stable performance on different data, and reproducibility under various conditions; and then contribute to targeted decisions, reduce the adverse effects induced by invasive diagnostic methods on LGGs patients."

We hope that the quality of paper is improved and can meet the quality requirements of Cancers for publication.